# *Entamoeba histolytica*: EhADH, an Alix Protein, Participates in Several Virulence Events through Its Different Domains

**DOI:** 10.3390/ijms25147609

**Published:** 2024-07-11

**Authors:** Dxinegueela Zanatta, Abigail Betanzos, Elisa Azuara-Liceaga, Sarita Montaño, Esther Orozco

**Affiliations:** 1Department of Infectomics and Molecular Pathogenesis, Center for Research and Advanced Studies of National Polytechnic Institute, Mexico City 07360, Mexico; dxinegueela.zanatta@cinvestav.mx; 2Postgraduate in Genomic Sciences, Autonomous University of Mexico City, Mexico City 03100, Mexico; elisa.azuara@uacm.edu.mx; 3Laboratory of Bioinformatics and Molecular Simulation, Faculty of Biological Chemistry Sciences, Autonomous University of Sinaloa, Sinaloa 80030, Mexico; mmontano@uas.edu.mx

**Keywords:** protozoan parasites, EhADH, ALIX family, scaffold protein, ESCRT machinery, intestinal and hepatic amoebiasis, epithelial cells

## Abstract

*Entamoeba histolytica* is the protozoan causative of human amoebiasis. The EhADH adhesin (687 aa) is a protein involved in tissue invasion, phagocytosis and host-cell lysis. EhADH adheres to the prey and follows its arrival to the multivesicular bodies. It is an accessory protein of the endosomal sorting complexes required for transport (ESCRT) machinery. Here, to study the role of different parts of EhADH during virulence events, we produced trophozoites overexpressing the three domains of EhADH, Bro1 (1–400 aa), Linker (246–446 aa) and Adh (444–687 aa) to evaluate their role in virulence. The TrophozBro1_1–400_ slightly increased adherence and phagocytosis, but these trophozoites showed a higher ability to destroy cell monolayers, augment the permeability of cultured epithelial cells and mouse colon, and produce more damage to hamster livers. The TrophozLinker_226–446_ also increased the virulence properties, but with lower effect than the TrophozBro1_1–400_. In addition, this fragment participates in cholesterol transport and GTPase binding. Interestingly, the TrophozAdh_444–687_ produced the highest effect on adherence and phagocytosis, but it poorly influenced the monolayers destruction; nevertheless, they augmented the colon and liver damage. To identify the protein partners of each domain, we used recombinant peptides. Pull-down assays and mass spectrometry showed that Bro1 domain interplays with EhADH, Gal/GalNAc lectin, EhCPs, ESCRT machinery components and cytoskeleton proteins. While EhADH, ubiquitin, EhRabB, EhNPC1 and EhHSP70 were associated to the Linker domain, and EhADH, EhHSP70, EhPrx and metabolic enzymes interacted to the Adh domain. The diverse protein association confirms that EhADH is a versatile molecule with multiple functions probably given by its capacity to form distinct molecular complexes.

## 1. Introduction

*Entamoeba histolytica* produces amoebiasis, a gastrointestinal infection that affects 5.81% of the world’s population and causes the death of more than 55,000 people each year [1]. This protozoan is generally transmitted by the fecal-oral route, and it colonizes and invades the intestine. In some cases, the intestinal infection becomes chronic and extraintestinal, producing mainly amoebic liver abscesses (ALA) [2,3,4].

During tissue invasion, the *E. histolytica* trophozoites adhere to lyse and phagocyte target cells [5]. Moreover, they can manage the oxidative stress and are capable of evading the host immune response [6]. The 124 kDa EhCPADH complex, constituted by the EhCP112 cysteine protease and the EhADH adhesin, is involved in these functions [7]. Furthermore, the complex also disrupts the epithelial barrier, contributing to tight junctions (TJs), adherens junctions (AJs) and desmosomes (DSMs) degradation [8,9]. It destroys the epithelia and synergizes the action of other parasite molecules during invasion, such as cysteine proteases (EhCPs) and prostaglandin E2 [10,11]. Moreover, the EhCPADH complex has been proposed as a vaccine candidate against *E. histolytica*, because hamsters immunized with a *pcDNA-Ehadh* and *pcDNA-Ehcp112* plasmids mixture presented protection against amoebiasis [12].

EhADH is a 75 kDa protein with two main regions: the N-terminus that contains the Bro1 domain (9–349 aa) [13,14], characteristic of the ALIX/AIP1 family proteins [15,16]; and the C-terminus, known as the V region for having a “V” structural morphology (395–657 aa) and where the EhADH adhesion motif is confined (480–600 aa) [13] (Figure 1). Previous studies have demonstrated that trophozoites overexpressing EhADH significantly increase their adhesion to target cells and the rate of erythrophagocytosis [14]. EhADH is also an accessory protein of the endosomal sorting complexes required for transport (ESCRT) machinery which interacts with the ESCRT components [17,18]. In addition, it associates with EhRabB, EhNPC1, EhNPC2, LBPA and actin, which together with the ESCRT machinery, participates in the vesicular trafficking and multivesicular bodies (MVBs) formation [17,19,20]. In brain pericytes, the human protein Alix forms a multiprotein complex with occludin and caveolin-1 TJs proteins [21], as EhADH does in epithelial cells, injuring the barrier functions.

Due to the EhADH’s seminal contribution to *E. histolytica* virulence, the aim of our work was to dissect the domains to facilitate their potential interactions with other molecules to form complexes involved in epithelial lysis and phagocytosis. The hypothesis of this study was that the multiple EhADH functions lie in its different domains, mainly in the Bro1 and the Adh domains. Hence, in this work we evaluated the in vitro and in vivo participation of three different EhADH domains (Bro1, Linker and Adh) in the distinct functions during the parasite virulence. The strategy followed here was the generation of trophozoites overexpressing the three respective regions, and to produce the recombinant proteins embracing each domain (Figure 2). Our results showed that the Bro1 domain enhances the cytopathic and cytotoxic effect and impairs the epithelial barrier of the mouse colon. The Linker segment moderately contributed to increase adhesion and the rate of erythrophagocytosis whereas the Adh domain produced the highest effect on the adhesion to and phagocytosis of the target cell, and it also augmented ALA formation. Then, we identified the diverse molecules that form complexes with EhADH and its domains to promote distinct functions during the virulence processes.

## 2. Results

### 2.1. EhADH Interacts with Proteins Involved in Vesicular Trafficking and Endocytosis

The EhADH adhesin is a 687 aa multifunctional molecule [7,8,10,11,12] that comprises a 3D structure mostly conformed by alpha helices [13], adopting a “high-heel shoe” form (Figure 2A). The protein exhibits a Bro1 domain with a boomerang figure and a V region shaped like the letter that contains the Adh domain. As far as we know, the Adh domain has not been found in other proteins of the ALIX family, giving the EhADH protein a particular characteristic among Alix proteins [13,14].

To execute its multiple functions, EhADH might directly or indirectly interact with other molecules [7,13,17,20] to form complexes. To gain more information on the putative EhADH partners during the virulence events, here, we first performed an in silico interactome analysis using the STRING server. The interactome displayed 10 EhADH partners (Figure 3A): EhCP112 (EHI_181230) [7], and another cysteine protease (EhCP, EHI_168240). Of note, EhADH participates in the formation of MVBs, so, it is expected that it also interacts with ESCRT machinery proteins. Effectively, the interactome revealed EhVps23 (EHI_178530; ESCRT-I) [17] and its orthologous TSG101 (EHI_131530) [22], EhVps32 (EHI_169820; ESCRT-III) [19], EhVps2 (EHI_194400; ESCRT-III) [19], and EhVps4-ATPase (EHI_118900; another ESCRT-accessory protein) [18]. In addition, EhADH interacts with actin (EHI_008780), a zinc-finger protein (EHI_117910), and a ubiquitin-protein ligase (EHI_010740) [22] (Figure 3A).

To experimentally verify some of these interactions, we produced the recombinant proteins GST-EhADH and GST to perform pull-down assays with *E. histolytica* lysates, followed by mass spectrometry. Most of the proteins that were revealed by the interactome were also identified in these experiments. The GST-EhADH-binging proteins with the highest score are shown in Table 1. Among the more relevant molecules appeared those involved in virulence-related functions, such as substrate degradation (EhCP112 and dipeptidyl peptidase III), vacuolar transport (EhVps23, EhVps36, EhVps22, EhVps32 and EhVps2), cholesterol trafficking (EhNPC1), cytoskeleton regulation (actin and actinin-like protein), calcium-binding proteins (EhGrainin 1, EhGrainin 2 and Eh-hand calcium-binding domain containing protein) and a zinc-finger protein. On the other hand, the control GST protein interacted with actin, small ribosomal subunit protein uS13, ubiquitin-protein ligase, and HECT-type E3 ubiquitin transferase (Appendix A).

Previously, some of the GST-EhADH interactions have been already suggested or tested by bioinformatics and experimental approaches [7,14,17,18,19,20,23,24]. Here, some of these associations were verified by pull-down and western blot assays, using specific antibodies for each protein (Figure 3B). These experiments demonstrated that EhADH forms protein complexes by direct or indirect binding to perform the distinct tasks in which it participates.

**Table 1 ijms-25-07609-t001:** Mass spectrometry analysis of *E. histolytica* proteins interacting with GST-EhADH.

Accession Number	Protein Name	Functions	References
EHI_181220 ^†^	EhADH *	Adhesion	[7,14]
EHI_181230	EhCP112 *	Substrate degradation	[7,25]
EHI_110750 ^†^	Dipeptidyl peptidase III	Substrate degradation	[22]
EHI_178530 ^†^	EhVps23 *	Vacuolar transport (ESCRT-I)	[17]
EHI_045320	EhVps36 *	Vacuolar transport (ESCRT-II)	[26]
EHI_131120 ^†^	EhVps22 *, ELL complex EAP30 subunit	Vacuolar transport (ESCRT-II)	[26]
EHI_169820 ^†^	EhVps32 *	Vacuolar transport (ESCRT-III)	
EHI_194400 ^†^	EhVps2 *	Vacuolar transport (ESCRT-III)	
EHI_118900 ^†^	EhVps4 *	ATP-binding and vacuolar transport (ESCRT-accessory protein)	[18]
EHI_080220 ^†^	EhNPC1 *	Cholesterol trafficking	[20]
EHI_008780 ^†^	Actin *	Cytoskeleton	[27,28,29]
EHI_164440 ^†^	Actinin-like protein	Cytoskeleton	[30,31]
EHI_108610 ^†^	Rab * family GTPase	Enzyme (signal transduction, vesicular trafficking, etc.)	[32,33,34,35]
EHI_167300 ^†^	EhGrainin 1	Calcium ion binding	[23,24,36]
EHI_111720 ^†^	EhGrainin 2	Calcium ion binding	[23,24,36]
EHI_197510 ^†^	EF-hand calcium-binding domain containing protein	Calcium ion binding	[36]
EHI_117910	Zinc-finger protein	Metal ion binding	[30,31]

The selected proteins had a reliability percentage greater than 95%. * Proteins have been published as related to virulence events. † Proteins with evidence of expression at membranes or phagosomes in trophozoites, according to AmoebaDB.

In particular, the in silico analysis of the interaction between EhADH and EhCP112 to form the EhCPADH complex, predicted that the 3D structure of both proteins is modified during Molecular Dynamics Simulation (MDS) and the binding is given through different regions and distinct amino acids (Figure 4).

### 2.2. Generation of Trophozoites Overexpressing EhADH Domains

To analyze the EhADH function, we divided the protein into three segments: (i) the Bro1 domain (1–400 aa); (ii) the central region of the protein (246–446 aa) called the Linker domain; and (iii) the adherence domain (444–687 aa), termed Adh domain (Figure 5A). We produced the Linker domain considering the last 103 aa of the Bro1 domain because it contains several putative phosphorylation residues that can modify the protein function, and the EhADH partners that could bind to these sites would not be detected due to the abundance of the Bro1 domain partners. The corresponding nucleotide sequences for each domain were synthetized, and then we cloned them into the *pTet* vector, generating the *pTet/Bro1*, *pTet/Linker* and *pTet/Adh* constructs. After sequencing, these plasmids were transfected into the parasites, generating the TrophozBro1_1–400_, TrophozLinker_246–446_ and TrophozAdh_444–687_ populations. The cellular growth of these populations was very similar (Appendix A).

At the same time, specific antibodies were produced against each fragment, derived from designed peptides with high specificity, exposed residues, and immunogenic regions (Figure 5A). The selected peptide sequences of EhADH domains were carefully screened to corroborate that they were exclusive of each fragment, and they do not cross-react with other *E. histolytica* proteins. Each sequence was conjugated to the KLH protein, and then, rats, mice and rabbits were immunized with Bro1, Linker and Adh peptides, respectively (Figure 5A). The antibodies’ specificities were probed on trophozoite lysates by western blot assays. As expected, the three antibodies recognized the endogenous EhADH protein (75 kDa) and the EhCPADH complex (124 kDa) (Figure 5B–D). Pre-immune sera did not detect any band.

Next, the α-Bro1, α-Linker and α-Adh antibodies were used to evaluate the expression of each domain in trophozoites transfected with tetracycline-induced plasmids. By western blot assays using lysates of transfected trophozoites, the antibodies detected the exogenous Bro1 (45 kDa), Linker (26 kDa) or Adh (31 kDa) transfected domains. The endogenous EhCPADH complex and EhADH protein were also revealed by the antibodies. However, in the TrophozControl, transfected with the empty *pTet* plasmid, the antibodies only recognized the endogenous EhCPADH complex and the EhADH protein (Figure 6A–C). The band corresponding to the complete EhADH protein (75 kDa) or the EhCPADH complex (124 kDa) migrated similarly in the TrophozControl and in the peptides-containing parasites. As a loading control, we used actin. The expression of the individual domains shows reduced intensity for EhADH for all three domains and in some cases, the EhCPADH complex was also altered, which may reflect an expression change in the protein turnover expression, consequently affecting the EhADH roles.

The EhADH protein is located in the plasma membrane and in the cytoplasm, mainly on endosomal membranes and phagosomes because it is a peripheral membrane protein [7,13,14]. Thus, we investigated the cellular fate of each one of the EhADH domains in the transfected trophozoites by confocal immunofluorescence assays, using the respective primary and secondary antibodies. An increase in fluorescence was observed in the TrophozBro1_1–400,_ TrophozLinker_246–446_ and TrophozAdh_444–687_, compared to the TrophozControl (Figure 6D–F). The TrophozBro1_1–400_ exhibited a more abundant staining in cytoplasmic dots that could correspond to vesicular structures, whereas in the TrophozLinker_246–446_, the pattern was more defined at the plasma membrane. Additionally, the TrophozAdh_444–687_ showed the mark in dots located in the cytoplasm and plasma membrane (Figure 6D–F).

So far, we generated transfected trophozoites overexpressing each of the EhADH domains, with a typical distribution at cytoplasmic vesicles and plasma membrane [13,14].

### 2.3. The Adh Domain Influences Adhesion and Phagocytosis by Trophozoites

EhADH participates in adhesion to and phagocytosis of erythrocytes [7,14] (see Figure 1). To study the contribution of each domain to those functions, we determined the adhesion efficiency to and the phagocytosis of red blood cells (RBCs) of the transfected trophozoites. For the adhesion experiments, cell mixtures were incubated at 4 °C for 5, 10, 15 and 30 min, as described in the Section 4. After 30 min incubation, transfected trophozoites overexpressing the Bro1 domain showed an increase in the RBCs adhesion (4.3 vs. 6.2 erythrocytes in TrophozControl vs. TrophozBro1_1–400_, respectively), but this increase was not statistically significant. TrophozLinker_246–446_ augmented the adhesion 62% more than the TrophozControl (7.0 vs. 4.3 RBCs in TrophozLinker_246–446_ vs. TrophozControl, respectively); and of note, the TrophozAdh_444–687_ exhibited the highest adhesion capacity, adhering 136% more than the TrophozControl (10.2 vs. 4.3 erythrocytes in TrophozAdh_444–687_ vs. TrophozControl, respectively), whose adhesion efficiency was given by the endogenous proteins (Figure 7A,B).

Then, we analyzed the rate of erythrophagocytosis of all populations incubating the trophozoites with RBCs at 37 °C. At 30 min, the TrophozBro1_1–400_ engulfed 28% more than the TrophozControl (1.3 vs. 1.0 erythrocytes in TrophozBro1_1–400_ vs. TrophozControl, respectively), but again it was not statistically significant. Instead, the TrophozLinker_226–446_ ingested 120% (2.3 vs. 1.0 erythrocytes in TrophozLinker_226–446_ vs. TrophozControl, respectively), and the TrophozAdh_444–687_ ingested 185% more than the TrophozControl (3.0 vs. 1.0 erythrocytes in TrophozAdh_444–687_ vs. TrophozControl, respectively) (Figure 7C,D). The TrophozAdh_444–687_ showed the highest adhesion efficiency and rate of erythrophagocytosis, maybe due to its enriched localization at the plasma membrane (see Figure 6F) or to the binding partners to this region, which makes them more exposed to catch erythrocytes. Given that the Linker domain contains several phosphorylation sites, it is probable that the phosphorylation states of the molecule influence the EhADH binding with other proteins, which would affect the adhesion and erythrophagocytosis events, but this is only a hypothesis that needs to be further experimentally tested.

### 2.4. The Bro1 Domain Enhances the Cytopathic and Cytotoxic Effect Produced by Trophozoites

In addition to phagocyte RBCs, bacteria and other cells and particles, the trophozoites are capable of destroying epithelia in vitro as well as in vivo. The EhADH protein is deeply involved in these events. However, we do not know the protein region involved in these processes. Therefore, we carried out cytopathic and cytotoxic assays with the transfected amoebae. Live trophozoites (for cytopathic effect) or lysates (for cytotoxic effect) were incubated with MDCK epithelial cell monolayers, and the cellular destruction produced by the parasites was quantified as described in the Section 4. Live trophozoites (2 × 10^5^) that overexpressed the Bro1 domain damaged MDCK monolayers up to 61% more than the TrophozControl (Figure 8A). Instead, the TrophozLinker_246–446_ and TrophozAdh_444–687_ produced similar epithelial monolayer destruction to the TrophozControl. Similar behavior was observed when trophozoite lysates were incubated with MDCK monolayers. Lysates from the TrophozBro1_1–400_ trophozoites destroyed the cells 47% more than the TrophozControl; unlike the TrophozLinker_246–446_ and TrophozAdh_444–687_, which presented similar results to the TrophozControl (Figure 8B). Thus, the TrophozBro1_1–400_ population enhanced the damage that trophozoites produce to epithelial cells, probably due to its capacity of interacting with other molecules forming complexes that participate in target cell destruction.

### 2.5. Bro1 and Adh Domains In Vitro Impair the Epithelial Barrier Functions

The first barrier to protect the epithelia is the TJ, which maintain the adjacent cells together, and regulate both ions and macromolecules flux by the paracellular pathway [37,38,39]. The EhADH directly affects epithelial junctions without the help of the EhCP112 or the intervention of other molecules. To investigate the active region of the EhADH that participates in the TJs opening, in this work, we used MDCK epithelial cell monolayers cultured in Transwell filters [40] incubated with trophozoites transfected with each one of the EhADH domains. In general, trophozoites overexpressing EhADH domains and control caused a decrease in the transepithelial electrical resistance (TEER), indicative of TJs opening. To quantify the damage, we took the TEER exhibited by the cell monolayers incubated with DMEM as 100% for 120 min at 37 °C. After incubation, the TrophozControl maintained 29% of TEER in MDCK cell monolayers. TrophozBro1_1–400_ consistently generated the greatest damage to MDCK cells, giving values of 13% TEER, whereas the cell monolayers incubated with the TrophozAdh_444–687_ and TrophozLinker_246–446_ exhibited 20% and 44% TEER, respectively (Figure 9A).

To investigate whether the EhADH domains also impact on the macromolecules’ paracellular permeability, we performed FITC-dextran flux assays [41]. The Ca^+^-chelator EDTA disassembles the intercellular junctions (IJs), permitting the free macromolecules flux; thus, cell monolayers incubated with EDTA were taken as 100% of dextran flux. After 120 min, all trophozoite populations increased the paracellular passage of macromolecules, compared to epithelial cells incubated only with DMEM. MDCK cells incubated with the TrophozControl permitted 41% FITC-dextran flow. Remarkably, the cell monolayers incubated with the TrophozBro1_1–400_ allowed 91% of macromolecule flux, similar to the EDTA-treated epithelial cells (Figure 9B). In concordance with ion flux results, the TrophozLinker_246–446_ and TrophozAdh_444–687_ trophozoites increased the macromolecule flux by 53% and 71%, respectively.

Once the EhADH protein impairs the TJs, it can reach AJs and desmosomal proteins, such as E-cadherin, β-catenin, desmoglein-2, and desmoplakin I/II, as it has been previously demonstrated. Thus, the breaking of the paracellular seal by EhADH allows *E. histolytica* to efficiently invade the epithelia.

### 2.6. The Bro1 Domain Injures the Mice Colonic Epithelium

So far, we have presented experimental evidence on the effect of the three EhADH regions on distinct in vitro virulence properties. To gain further insight on the in vivo participation of the EhADH domains, we investigated the effect of the TrophozBro1_1–400_, TrophozLinker_246–446_ and TrophozAdh_444–687_ on the C57/BL6 mice gut epithelia [42]. For these experiments, we rectally injected 10^5^ trophozoites in anesthetized mice. After 30 min of incubation, Evan’s blue tracer was also rectally injected to determine tissue damage by the tracer absorption in the colon [9,20,43]. Mice were euthanized, colons were removed, and the tracer was eluted and spectrophotometrically measured at OD_610nm_. Mice treated only with PBS showed no evidence of harm along the colon (OD_610nm_ = 1.44), whereas mice inoculated with the TrophozControl presented visible damage (OD_610nm_ = 2.11) (Figure 9C,D). The epithelia from all mice inoculated with TrophozBro1_1–400_, TrophozLinker_246–446_ and TrophozAdh_444–687_, displayed greater penetration of Evan’s blue with the naked eye. Remarkably, after tracer elution and quantification by densitometric analysis, we found that the TrophozBro1_1–400_ produced 159% (OD_610nm_ = 5.47) more damage than the TrophozControl, followed by the TrophozAdh_444–687_ and TrophozLinker_246–446_ that induced 105% (OD_610nm_ = 4.35) and 38% (OD_610nm_ = 2.93), respectively. These findings indicate that trophozoites overexpressing the Bro1 domain produced the highest permeability of colonic epithelium, probably because this region interacts with other parasite molecules that contribute to disrupt the IJs or associates with epithelial proteins to mimic the homo- and heterodimeric interactions to open TJs.

These results revealed that the Bro1 domain may be directly involved in the impairment of the epithelial barrier integrity produced by trophozoites, as it was strongly suggested by the in vitro (see Figure 8 and Figure 9A,B) and in vivo strategies used here. We hypothesize that it may be due to the ability of the Bro1 domain to interact with several molecules, forming complexes that induce the epithelial damage. The Bro1 domain’s interactions with other proteins to perform distinct cellular functions have been described for other ALIX family members [15,16].

### 2.7. Trophozoites Overexpressing the Adh Domain Produced Larger Liver Abscesses

Generally, *E. histolytica* primarily damages the intestine; however, eventually it migrates to the liver producing ALA that can be lethal [44,45]. It is unknown whether the same virulence molecules and mechanisms act in both cases. To explore the role of the EhADH domains in ALA formation, we intraportally inoculated TrophozControl and trophozoites overexpressing the distinct EhADH domains in anesthetized hamsters. After seven days, animals were anesthetized again, and livers were extracted to examine the damage. All infected animals presented augmented weight livers, more than twice the size of the liver from non-infected hamsters (Figure 10A,B). Surprisingly, the abscesses produced by the distinct trophozoite populations consistently exhibited morphological differences (Figure 10B). TrophozControl parasites produced small and dispersed abscesses along the tissue, whereas TrophozBro1_1–400_ caused localized damage in one of the lobes and the abscesses appeared granular. Moreover, the coloration of the liver was lighter than the controls, probably due to the interference with the blood supply to the organ (Figure 10B). The TrophozLinker_246–446_ provoked more significant damage in the central area of the liver, and abscesses appeared more compact and uniform. TrophozAdh_444–687_ produced ALA macroscopically similar to the TrophozControl; however, a larger number of abscesses were formed; in addition, a dark or purplish coloration was observed in some liver areas that could be due to necrosis (Figure 10B). Quantification of injured hepatic tissue revealed that animals inoculated with the TrophozControl presented around 37.5% damaged tissue, but TrophozAdh_444–687_ provoked the highest injury to the liver (72.5%), followed by TrophozBro1_1–400_ and TrophozLinker_246–446_ (57% and 49.5%, respectively) (Figure 10C).

The damage produced by TrophozAdh_444–687_ could be related to the increase in adhesion and phagocytosis that these parasites exhibited, as it was shown in other experiments of this work (see Figure 7), or to the formation of protein clusters obstructing the blood flux and concentering harmful molecules derivative from the parasites and the host.

### 2.8. The Bro1, Linker and Adh Recombinant Domains Interact with Virulence-Related and Cytoskeleton Proteins

The fact that each domain differentially contributes on distinct *E. histolytica* virulence properties does not mean that the domain alone could be able to perform the function. It is more likely that each domain recruits other proteins to form functional complexes, which together play a role in virulence.

To explore the molecular mechanisms and unveil some of the proteins interacting with each EhADH domain, we generated His-Bro1 (47 kDa), GST-Linker (50 kDa) and GST-Adh (56 kDa) recombinant domains and we checked their identity by western blot, using the respective specific antibodies (Figure 11A). Then, we performed pull-down assays using these recombinant proteins that were incubated with trophozoite lysates. Mass spectrometry results showed that His-Bro1 protein interacted with Gal/GalNAc lectin; EhCP112 and EhCP2; ESCRT machinery proteins (ESCRT-I and ESCRT-III); EhVps26 (a component of the retromer complex); actin and tubulin (Table 2). Randomly selected proteins from these interactions were verified by western blot assays, corroborating that the His-Bro1 protein associates with EhCP112, Gal/GalNAc lectin, EhVps23 (ESCRT-I), EhVps32 (ESCRT-III) and actin (Figure 11B). Remarkably, this domain is also associated with the full-length EhADH protein (75 kDa) and with the EhCPADH (124 kDa) complex.

The recombinant Linker protein (GST-Linker) also interacted with the full-length EhADH protein; EhCP112; ubiquitin; GTP-binding proteins, such as Rabs; EhVps35 (a component of the retromer complex); EhNPC1 (a cholesterol transporter); and the EhHSP70 chaperone, among others (Table 3). Western blot experiments corroborated the association of the Linker domain to EhCP112, ubiquitin, EhRabB, EhNPC1 and EhHSP70 (Figure 11C). Like the Bro1 domain, the GST-Linker protein binds to the full-length EhADH and EhCPADH complex.

Finally, the Adh domain (GST-Adh) interacted with the full-length EhADH; the EhHSP70 chaperone; several enzymes involved in metabolic processes; ribosomal proteins; and a 20 kDa antigen (Table 4). The GST-Adh binding to the full-length EhADH protein, EhCPADH complex, EhCP112, EhHSP70 and EhPrx was verified by western blot experiments in the pull-down assays eluted (Figure 11D).

The His and GST proteins were included as controls and in the mass spectrometry analysis different interacting proteins to His-Bro1, GST-Linker and GST-Adh were obtained (Appendix A). The exception was the binding of His and His-Bro1 with one of the actin proteins (accession number: EHI_182900).

In conclusion, these findings obtained using in vitro virulence assays, two different animal models, recombinant protein, mass spectrometry and others, showed that EhADH is indeed a versatile scaffold protein that interacts with several molecules, many of which have been identified as virulence factors. The EhADH domains differentially participate in adhesion, phagocytosis, epithelial injury, and ALA formation, through their interactions with proteins involved in these events, such as adhesins, molecules relevant during MVBs formation, retrograde transport proteins, cytoskeleton molecules, vesicular trafficking proteins, substrate degradation and metabolic enzymes, and heat-shock proteins. EhADH also binds to several host molecules during epithelial invasion facilitating entrance, colonization and injury [8].

## 3. Discussion

EhADH belongs to the ALIX/AIP1 family since it has the Bro1 domain at its N-terminus, which is a signature of this family [13,14]. The ALIX family members are involved in cell adherence, endosomal protein sorting, cytoskeletal remodeling, pH regulation and apoptosis, among other functions [15,16]. The versatility of all these functions, has been reported for other protozoa, such as *Dictyostelium discoideum*, where Alix has been identified and involved in the developmental arrest of the Alix null mutant [58]. Furthermore, in the parasite *Echinococcus multilocularis*, the Alix protein has been detected as an important marker of extracellular vesicles, together with TSG101 and tetraspanins [59]. This and other functions have also been attributed to EhADH in *E. histolytica* trophozoites [18,60,61,62,63,64,65,66], conducted by us to carefully explore the EhADH structural and functional characteristics. To achieve this study, we developed two general strategies: (i) The analysis of the complete EhADH protein (1–687 aa) and the generation of three fragments encompassing the amino (Bro1: 1–400 aa), the middle (Linker: 246–446 aa), and the carboxy-terminus (Adh: 444–687 aa) regions; the latter containing the adherence domain. These segments were overexpressed in trophozoites to separately analyze their influence on different in vitro and in vivo events. (ii) The production of recombinant proteins using the mentioned segments, to identify *E. histolytica* proteins interacting with each one of them. In addition to EhADH, the *E. histolytica* genome has two other ALIX family members: EhADH-like 1 and EhADH-like 2. However, they differ from EhADH in their sequence. Furthermore, they do not contain the Adh domain, which is fundamental for certain parasite virulence functions [14]. As a tool, we generated specific antibodies against each one of the domains, using unique amino acid sequences that were not found in the other segments nor in other amoebic proteins. The three antibodies recognized also the EhCPADH complex (formed by EhADH and EhCP112 protein) and the EhADH protein, proving the fragment identity. Interestingly, the trophozoites transfected with each one of the segments examined with the antibodies gave distinct fluorescence patterns and the recombinant peptides bound different proteins. The antibody against Bro1 domain displayed profusely illuminated vesicles of different size, possibly corresponding to endosomes or MVBs, according to previous reports indicating that Alix proteins have been located on the endosomal membranes [15], as in *E. histolytica* trophozoites [13,14]. The antibody against the Linker domain reacted with regions of the plasma membrane. It may be due to its association with the whole EhADH, as it was revealed by the mass spectrometry analysis (Table 3). Moreover, previous reports revealed EhADH in the plasmatic membrane [15,16,67]. The antibody against the adherence domain stained many vesicles on the trophozoites surface, suggesting that this region participates in the secretion process in which the EhADH is also involved.

The impact of the EhADH domains on different in vitro virulence properties was also investigated using the transfected trophozoites. Remarkably, the three different trophozoite populations behaved differentially during these experiments. The trophozoites transfected with the Adh domain exhibited both a higher adherence efficiency and rate of phagocytosis than the TrophozControl, whereas TrophozBro1_1–400_ showed a lower effect on these events. It is probable that the presence of a higher amount of the Adh domain in the peripheral vacuoles observed in the fluorescence images exposed the trophozoites to a higher number of RBCs which facilitated their ingestion.

One of the first signs of epithelial damage produced by the trophozoites is the IJs aperture, which produces separation of the cells from the epithelia. The purified EhADH recombinant protein also produced the same effect. To explain this process, we previously showed that EhADH binds to claudin-1, delocalizing it and other TJs proteins and first producing the TJs opening, and later the AJs and DSMs rupture. Furthermore, the purified EhADH is capable of binding to the epithelial host clathrin and caveolin. In this paper, we have found that TrophozBro1_1–400_ and their lysates displayed the highest capacity to destroy epithelial cell monolayers. While the TrophozLinker_246–446_ and TrophozAdh_444–687_ exhibited a minor effect compared to the TrophozControl. In agreement with these results, the trophozoites transfected with the Bro1 and Adh domains were more efficient than the TrophozControl to open the IJs, provoking a dramatic TEER drop. In concordance, these populations also permitted a higher dextran flux than the TrophozControl, indicating that the macromolecules flux in the epithelia was also altered. These experiments proved that the EhADH carboxy terminus, carrying the adherence domain, is also involved in the IJs rupture. We hypothesized that EhADH is a highly dynamic protein, capable of interacting with several proteins and through distinct regions with the same protein, as has been described for other Alix proteins, which are capable of forming multimers using its V domain [15,68,69,70].

The experiments performed with animal models also showed the distinct capacities of the EhADH domains to injure tissues. In addition, they confirmed that at least certain events that occur in vitro also occur in vivo, validating the in vitro models to study the *E. histolytica* virulence. EhADH establishes several contacts that allow it to act as a scaffold and participate in different functions. In this project, it was shown that EhADH interacts with ubiquitin, and it is believed that ubiquitination could regulate the dimerization of Alix proteins through its V domain, as occurs with other members of this family [71]. EhADH recruits not only amebic molecules but also host proteins during the tissue invasion, such as cytokines and molecules from neutrophils and other cells that congregate in the epithelia to carry out their defensive task, but paradoxically also damaging the host. In fact, we have reported the interaction of EhADH with IJs proteins [8]. In addition, metabolic enzymes and other proteins as EhADH partners suggest the participation of this adhesin in other functions apparently not related to invasion. Moreover, the limitations of these experimental approaches must be considered. For example, it is plausible that steric hinderances of domains in the whole molecule might prevent the binding of potential partners that can bind to each domain when they are outside of the molecule. However, the experiments presented in this article give an approach to the interactions carried out by EhADH.

Taking into consideration all these results, they first showed the cooperative effect of three distinct EhADH regions to interact with a target cell. While a part of the protein participates more active in adhesion, another part produces the tissue cell separation. Second, molecular cooperation with other proteins is also necessary to carry out the multiple functions of this scaffold protein. Our findings showed the participation of diverse molecules in adhesion-phagocytosis events and in the epithelial destruction and ALA formation, by the association to different molecules. The EhADH dynamics and versatility help it to interact with different molecules to conduct them to the site that the trophozoites require (Figure 12). Interestingly, it is capable of joining to the same molecule by different regions depending on its 3D structure, as can be seen in Figure 4, where we have shown the interaction of EhADH with EhCP112 at different molecular dynamic times.

## 4. Materials and Methods

### 4.1. Chemical and Reagents

The quality of chemical reagents and substances met specific standards of high purity and reliability. All reagents employed in this work underwent rigorous quality control measures to ensure they were free from impurities that could introduce variability or bias into the experimental assays.

### 4.2. Generation of Secondary and Tertiary Structure of EhADH and Its Bro1, Linker and Adh Domains

The EhADH (accession number: EHI_181220) amino acid sequence was retrieved from the AmoebaDB [30]. By in silico analysis, the Bro1 domain, characteristic of ALIX family proteins, was located and aligned to the complete EhADH amino acid sequence and compared with ALIX family orthologous (Protein BLAST and SMART). Other EhADH domains were identified by UniProt and ScanProsite servers [31]. The primary structure was represented using the Illustrator for Biological Sequences (IBS) server.

The predicted EhADH 3D structure was obtained using the Alix and Bro1 crystal proteins from *Homo sapiens* (PDB: 2OEV) and *Saccharomyces cerevisiae* (PDB: 1ZB1), respectively. The model obtained from the I-TASSER server [72] with the higher C-score was selected and the 3D structure was visualized using the UCSF Chimera software v1.16. The interactome was performed using EhADH as a bait and the STRING database.

### 4.3. Molecular Dynamics Simulations (MDS)

The 3D structures of EhADH and EhCP112 proteins were obtained from previous reports [13]. Then, models were submitted to 500 ns of MDS and their validation was performed as previously reported [13], in the Hybrid Cluster Xiuhcoatl of the Center for Research and Advanced Studies of the National Polytechnic Institute (CINVESTAV-IPN), Mexico. The protein-protein docking studies were carried out in the ClusPro server and protein visualization was performed by the Visual Molecular Dynamics (VMD) program.

### 4.4. Production of α-Bro1, α-Linker and α-Adh Antibodies

To generate specific antibodies for each one of the EhADH domains, three specific peptides (Bro1: 212-NQLIPSVDAFKTFYKITV-229 aa; Linker: 426-IEAETTASFEKGITALDS-443 aa; and Adh: 495-KFRQFENDIKLLCEGNIQ-512 aa; Figure 3A) were commercially synthesized together with the KLH (Keyhole Limpet Hemocyanin) tag to increase their immunogenicity (GenScript; Shangai, China). Rats (Wistar, n = 3), mice (Balb/cJ, n = 7) and rabbit (New Zealand; n = 1) were immunized with Bro1 (200 µg), Linker (100 µg) or Adh (600 µg) peptides resuspended in TiterMax^®^Gold (Sigma; Burlington, MA, USA) adjuvant (1:1), respectively. Three more immunizations were performed at 15-day intervals, followed by bleeding to obtain the antibodies. Pre-immune sera were obtained before immunizations.

### 4.5. Trophozoites and Epithelial Cell Cultures

Trophozoites of *E. histolytica* strain HM1:IMSS, clone A were axenically cultured in TYI-S-33 medium at 37 °C. Parasites were harvested during the logarithmic growth phase by chilling the culture tubes for 10 min in an ice-water bath and collected by centrifugation at 360× *g* for 5 min [73].

Madin Darby canine kidney (MDCK) epithelial cells type I [74] were grown in DMEM medium (Gibco; Waltham, MA, USA) supplemented with 100 IU/mL penicillin (in vitro), 100 mg/mL streptomycin (in vitro), 10% fetal bovine serum (FBS; Gibco), and 0.08 U/mL insulin (Eli Lilly; Mexico City, Mexico), at 37 °C in a 95% air and 5% CO_2_ atmosphere.

### 4.6. Cloning of the EhADH Domains

*Ehadh* gene encoding was divided in three fragments: Bro1 (nt 1–1200), Linker (nt 738–1338) and Adh (nt 1332–2061) domains that were PCR-amplified, using specific primers (Appendix A, transfected trophozoites), 10 mM dNTPs, 100 ng of *E. histolytica* cDNA as template and 2.5 U of Taq DNA polymerase (Gibco). Oligonucleotides used in this work specifically recognized sequences present in EhADH protein but not in those encoded by the other *Ehadh*-like genes [14] or other *E. histolytica* proteins. PCR assays were carried out for 35 cycles (30 s at 95 °C, 30 s at 60 °C, and 1.5 min at 72 °C). The PCR-amplified products were cloned into the vector *pJET1.2/blunt*. Next, they were cloned into the KpnI and BamHI restriction sites of the *pEhHYG-tetR-O-CAT* (*pTet*) plasmid, which contains the tetracycline (Tet) resistance operon of Escherichia coli and hygromycin resistance (Hyg) conferring gene, as selectable markers [75], producing the *pTet/Bro1*, *pTet/Linker* and *pTet/Adh* constructs. *E. coli* DH5α bacteria were transformed with the plasmids, that were purified by affinity columns using the QIAGEN Maxi kit (QIAGEN; Venlo, The Netherlands) and automatically sequenced to corroborate the error-free sequences.

### 4.7. Generation of TrophozBro1_1–400_, TrophozLinker_246–446_ and TrophozAdh_444–687_ Parasites by Transfection

*pTet*, *pTet/Bro1*, *pTet/Linker* and *pTet/Adh* plasmids (200 μg) were transfected by electroporation [76] into exponentially growing trophozoites, generating TrophozControl, TrophozBro1_1–400_, TrophozLinker_246–446_ and TrophozAdh_444–687_ populations, respectively. To select the transfected trophozoites, medium was supplemented with 3–5 mg/mL geneticin (G-418; Gibco), a hygromycin derivative. To induce the expression of different EhADH domains, 1 µg/mL tetracycline (Sigma-Aldrich; Burlington, MA, USA) was added to the TYI-S-33 medium.

### 4.8. Expression and Purification of GST-EhADH, His-Bro1, GST-Linker and GST-Adh Recombinant Proteins

The full-length *Ehadh* gene and the sequence of the three fragments encoding for the Bro1, Linker and Adh domains were PCR-amplified, using specific primers (Appendix A, recombinant proteins), using 10 mM dNTPs, 100 ng of *E. histolytica* cDNA as template and 2.5 U of Taq DNA polymerase (Gibco). The PCR assay was carried out for 35 cycles (30 s at 95 °C, 30 s at 60 °C, and 2 min at 72 °C) and the PCR-amplified products were cloned in the positive selection vector *pJET1.2/blunt*. Next, *Ehadh*, *Linker* and *Adh* sequences were cloned in *pGEX-6P-1* plasmid (GST tag), whereas the *Bro1* sequence was cloned in *pColdIDNA* plasmid (His tag), using the corresponding restriction sites (Appendix A). Plasmids were purified by affinity columns (QIAGEN) and automatically sequenced to corroborate the error-free sequences.

*E. coli* C43 (DE3) bacteria were transformed with the *pGEX/Ehadh* plasmid to induce the GST-EhADH expression with 1 mM IPTG. *E. coli* Rosetta (DE3) bacteria were transformed with the *pCold/Bro1* plasmid to produce the His-Bro1 recombinant protein. *E. coli* BL21 (DE3) bacteria were transformed with *pGEX/Linker* and *pGEX/Adh* plasmids to generate GST-Linker and GST-Adh recombinant proteins. Bacteria were lysed with 2% sarcosyl and 0.5% Triton X-100 in PBS and sonicated at 4 °C. The His-Bro1 recombinant protein was purified by cobalt beads in an imidazole gradient, while GST-EhADH, GST-Linker and GST-Adh proteins were purified by glutathione-Sepharose beads, following the manufacturers’ instructions.

### 4.9. Western Blot Experiments

Trophozoites lysates (35 µg) or purified recombinant proteins (35 µg) were electrophoresed in 10% and 12% sodium dodecyl sulfate-polyacrylamide gels (SDS-PAGE) and transferred to nitrocellulose membranes. Membranes were overnight (ON) incubated with the different primary antibodies: mouse α-His (1:500), goat α-GST (1:10,000), rat α-Bro1 (1:2000), mouse α-Linker (1:1000), rabbit α-Adh (1:2000), rabbit α-EhCPADH (1:35,000), rabbit α-EhCP112 (1:5000), rat α-EhVps23 (1:500), mouse α-EhVps32 (1:500), mouse α-actin (1:3000; kindly donated by Dr. José Manuel Hernández from Cell Biology Department, CINVESTAV-IPN; Mexico City, Mexico), mouse α-ubiquitin (1:100), rabbit α-EhRabB (1:500), rabbit α-Gal/GalNAc lectin (1:500; kindly donated by Dr. William A. Petri Jr from the Department of Pathology, University of Virginia, USA), rabbit α-EhNPC1 (1:3000), rabbit α-HSP70 (1:1000; [77]) and mouse α-EhPrx (1:500; kindly donated by Dr. José Luis Rosales Encina from the Infectomics and Molecular Pathogenesis Department, CINVESTAV-IPN). Membranes were washed, and incubated with the corresponding α-mouse, α-goat, α-rat or α-rabbit HRP-labelled secondary antibody (1:10,000; Sigma), and then revealed with ECL Prime detection reagent (GE-Healthcare; Chicago, IL, USA), according to the manufacturers’ instructions on a MicroChemi System (DNR Bio-Imaging).

### 4.10. Pull-Down and Mass Spectrometry Analyses

The His-Bro1 recombinant protein was purified by cobalt beads in an imidazole gradient, while GST-EhADH, GST-Linker and GST-Adh proteins were purified by glutathione-Sepharose beads. Subsequently, resin-bound proteins (200 μg) were incubated with total trophozoite lysates (2 mg) for 4 h at 4 °C with gentle agitation. After this, three washes with elution buffer were performed. The samples were centrifuged, and the supernatant was recovered. The proteins were separated by SDS-PAGE and visualized by silver stain (Bio-Rad; Hercules, CA, USA). As controls for these experiments, the pull-down assay was carried out only with the resins.

The parasite proteins were cut from the stained gel and processed by mass spectrometry on nano UPLC ACQUITY “M” Class coupled with a QToF Synapt G2-Si mass spectrometer (Waters Corporation; Milford, MA, USA) in the Genomics, Proteomics and Metabolomics Unit from CINVESTAV-IPN. The resulting information was analyzed on the Protein Lynx Global Server (PLGS) v3.0.3 (Waters Corporation). The data obtained were cured and selected as their associations when they had a reliability percentage greater than 95%. The reliability percentage was obtained during the peptide identification process. The software ensured that each identified ion meets the physicochemical parameters chosen. To test the specificity of the associations, we performed the pull-down assay with the total amoebic extracts and the resins but without the recombinant protein of interest.

### 4.11. Immunofluorescence Assays

Transfected trophozoites were grown on coverslips, fixed with 4% paraformaldehyde at 37 °C for 1 h, permeabilized with 0.5% Triton X-100 and blocked with 10% FBS in PBS (140 mM NaCl, 2.7 mM KCl, 10 mM Na_2_HPO_4_, 1.8 mM KH2PO4, pH 7.4). Then, cells were incubated ON at 4 °C with α-Bro1 (1:100) α-Linker (1:100) or α-Adh (1:100) antibodies. After extensive washing, samples were incubated for 30 min at 37 °C with the corresponding α-rat, α-mouse or α-rabbit FITC-labelled secondary antibody (1:100). Nuclei were stained with 4′,6-diamidino-2-phenylindole (DAPI) and fluorescence was preserved using Vectashield antifade reagent (Vector; Torrance, CA, USA). Preparations were examined through a Carl Zeiss LMS 700 confocal microscope, in 0.5 µm laser sections and processed with ZEN 2009 Light Edition Software Edition 5.5 (Zeiss; Jena, Germany).

### 4.12. In Vitro Virulence of E. histolytica

Trophozoites were incubated at 4 °C or 37 °C with erythrocytes (1:25 ratio) at different times for adhesion or phagocytosis assays, respectively [7]. For erythrophagocytosis experiments, adhered and non-ingested erythrocytes were lysed by incubation in distilled water for 10 min at room temperature (RT). Later, trophozoites and ingested erythrocytes were lysed using absolute formic acid and the hemoglobin was quantified by spectrophotometry at λ = 400 nm [78]. In some experiments, adhered and ingested erythrocytes were contrasted by Novikoff staining [79] and samples were observed through a light microscope (Axiolab, Zeiss).

For cytopathic assays, trophozoites (50, 100, and 200 × 10^3^) were incubated with confluent MDCK cell monolayers (1:20 ratio) for 2 h. After this time, the reaction was stopped by cooling cell culture plates in an ice-water bath and after several washes with cold PBS the adhered trophozoites were released.

For cytotoxic assays, trophozoites (50, 100, and 200 × 10^3^) were washed twice with ice-cold PBS and lysed by freeze-thawing in the presence of 100 mM phydroxymercuribenzoate (PHMB) and 40 μg/mL of E-64. Trophozoite lysates were incubated with confluent MDCK cell monolayers (1:20 ratio) for 2 h.

Later, the remaining epithelial cells were fixed with 2.5% glutaraldehyde and stained with 1% methylene blue for 10 min. The MDCK cells’ destruction was represented compared to control cells not incubated with trophozoites. Dye concentration was determined spectrophotometrically at λ = 660 nm [7].

### 4.13. Measurement of Transepithelial Electrical Resistance (TEER)

MDCK cells were seeded on Transwell filter supports (6.5 mm diameter and 0.4 μm pore; Corning; Corning, NY, USA) [40]. Three days after plating, and after confirming through an inverted microscope that monolayers have reached confluency, MDCK cells were apically incubated with live trophozoites (10^5^/cm^2^), and TEER was monitored for 120 min. The TEER was measured using an EVOM epithelial voltmeter (World Precision Instruments; Sarasota, FL, USA). TEER values were obtained by subtracting cell-free filter readings.

### 4.14. Paracellular Flux Assays

FITC-dextran (3 mg/mL) of 4 kDa (Sigma) was added to the apical side of confluent MDCK epithelial cells incubated with live trophozoites and seeded in Transwell filter supports [41]. After 120 min incubation at 37 °C with gentle shaking in darkness, samples from the basal chamber were collected and the diffused fluorescent tracer was measured in a fluorimeter (excitation λ = 547 nm; emission λ = 572 nm). Emission values were converted to FITC-dextran concentration, using a standard curve [9]. As a positive control, before tracer addition, cells were incubated with 5 mM EDTA.

### 4.15. Permeability Experiments In Vivo

Evan’s blue-based in vivo colon permeability assays were performed as described [43]. Briefly, pathogen-free C57/BL6 male mice (6–8 weeks old, ∼25 g, n = 3) were intraperitoneally anesthetized with 125 mg/kg of ketamine hydrochloride (Sanofi; Gentilly, France) and 12.5 mg/kg of xylazine (Phoenix Scientific; Bangkok, Thailand) and then rectally inoculated with live trophozoites (10^6^ cells in 200 μL of TYI-S-33 medium) for 30 min. After laparotomy, a 22G polyethylene tube was inserted into the colon, adjacent to the cecum, and ligated. Then, the remaining stool was rinsed out, and 1 mL of 1.5% Evan’s blue dye (Sigma-Aldrich) was instilled for 15 min. After washing with PBS until the perianal washout was clear, animals were euthanized, and the colon extracted. The colon was longitudinally opened and rinsed again with PBS, followed by 1 mL of 6 mM N-acetylcysteine to remove dye within the mucus. Colons were incubated ON in 2 mL of formamide at RT and the extracted dye was spectrophotometrically measured at λ = 610 nm. Values were expressed as arbitrary units per gram of tissue.

### 4.16. ALA Formation

Six week-old male hamsters (*Mesocricetus auratus*) weighing 70 ± 5 g (n = 5), were fasted for 24 h prior to surgery. Subsequently, they were anesthetized with 3% isoflurane (PiSA; Mexico City, Mexico) and a longitudinal incision of the abdominal wall was made to expose the port vein and livers. Live trophozoites (2 × 10^6^ in 200 mL of TYI-S-33 medium) were intraportally inoculated into the animals. Eight days after the challenge, hamsters were sacrificed with an overdose of anesthetic. The whole liver was weighed, and then the liver lesion was dissected and weighed to calculate the percentage of the damaged tissue in relation to the total liver weight.

### 4.17. Statistical Analysis

All data in this work followed a normal distribution probed by W, skewness, skewness shape, excess kurtosis, kurtosis shape and outliers’ values (Appendix A). Values were displayed as mean and standard error. Statistical analyses were performed by Student’s *t* test and two-way ANOVA, using GraphPad Prism 8.0 software. The scores showing statistically significant differences are indicated with asterisks in the graphs and assumed when * *p* < 0.05, ** *p* < 0.01, *** *p* < 0.001 or **** *p* < 0.0001.

### 4.18. Ethics Statement

The CINVESTAV-IPN fulfilled the standard of the Mexican Official Norm (NOM-062-ZOO-1999) and Technical Specifications for the Care and Use of Laboratory Animals based on the Guide for the Care and Use of Laboratory Animals, 2011, NRC, USA with the Federal Register Number BOO.02.03.02.01.908. This was awarded by the National Health Service, Food Safety and Quality (SENASICA) belonging to the Animal Health Office of the Secretary of Agriculture, Livestock, Rural Development, Fisheries and Food (SAGARPA), an organization that verifies the state compliance of such Mexican Official Norm (NOM) in Mexico. The Institutional Animal Care and Use Committee (IACUC/ethics committee) of CINVESTAV, as the regulatory office for the approval of research protocols involving the use of laboratory animals and, in fulfilment of the NOM, has reviewed and approved all animal experiments (Protocol Number 0225-16, CICUAL 001). The study is reported in accordance with the ARRIVE guidelines. In addition, all methods were performed in accordance with the relevant guidelines and regulations.

## 5. Conclusions

The study of the EhADH and its domains confirmed that EhADH is a multifunctional protein that differentially participates in several virulence events due to its domain’s versatility, allowing the interaction with molecules involved in adhesion, phagocytosis, lysis and epithelial colonization. We are convinced that this molecule can be used as a potential candidate to develop therapeutical approaches or vaccines in the future.

## Figures and Tables

**Figure 1 ijms-25-07609-f001:**
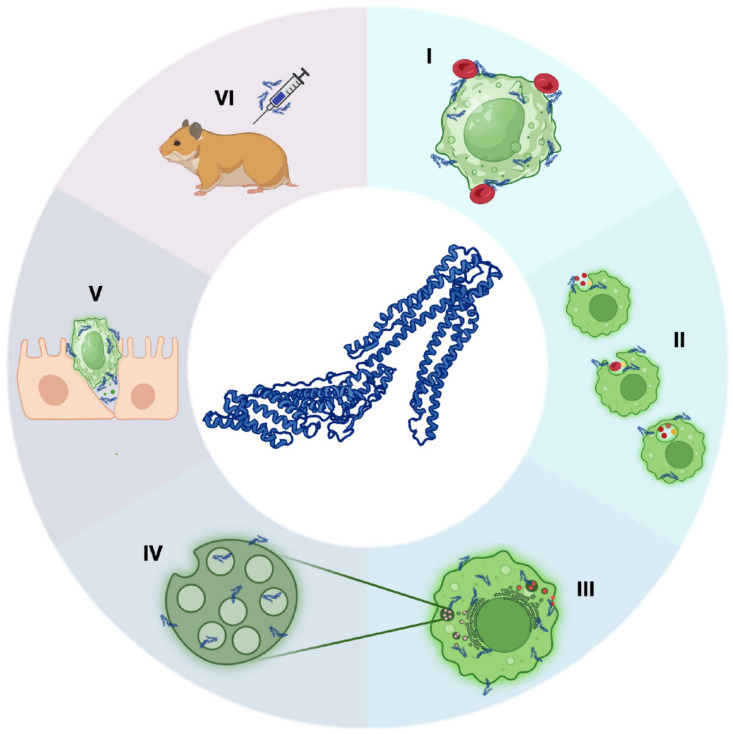
Scheme of the EhADH functions. (**I**) Adherence to target cells [7,14]. (**II**) Phagocytosis [7,14]. (**III**,**IV**) Vesicular trafficking and MVBs formation (it participates as an accessory protein of the ESCRT machinery) [14]. (**V**) Epithelial barrier impairing [7]. (**VI**) Potential vaccine candidate against amoebiasis [12]. In the center: 3D model of EhADH.

**Figure 2 ijms-25-07609-f002:**
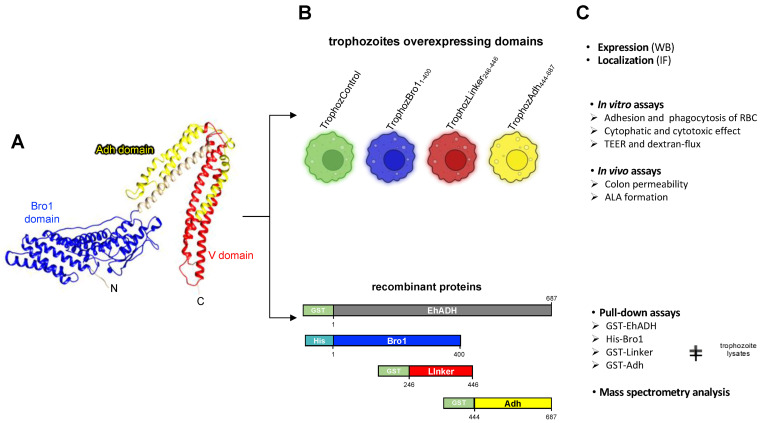
Experimental approaches to study the EhADH domains. (**A**) EhADH 3D structure obtained from the I-TASSER server, showing in distinct colors the Bro1 (blue) and the V (red) domains as well as the Adh domain (yellow). N: amino terminal. C: carboxy terminal. (**B**) Upper panel: Generation of trophozoites overexpressing Bro1 (TrophozBro1_1–400_), Linker (TrophozLinker_246–446_) and Adh (TrophozAdh_444–687_) domains. Trophozoites transfected with the empty vector (*pTet*) are used as control (TrophozControl). Lower panel: Production of recombinant proteins with different tags containing the whole EhADH proteins or embracing different domains. (**C**) Upper panel: Experiments carried out to analyze the expression and localization of transfected trophozoites. Middle panel: Trophozoites overexpressing different domains were used for in vitro and in vivo virulence assays. Lower panel: Identification of binding-partners of each EhAdh domain by pull-down assays and mass spectrometry analysis.

**Figure 3 ijms-25-07609-f003:**
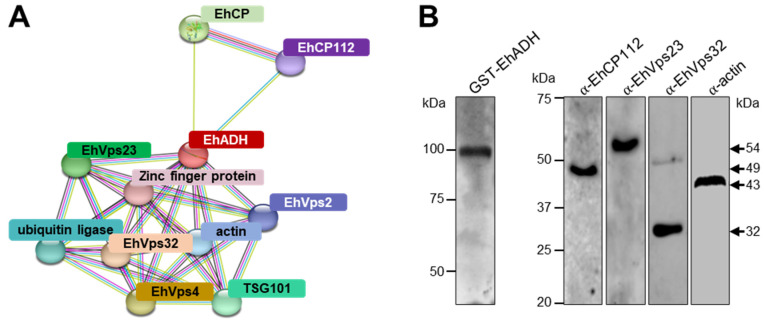
EhADH interactions with other *E. histolytica* proteins. (**A**) EhADH interactome generated by STRING. (**B**) Pull-down assays using the GST-EhADH recombinant protein and *E. histolytica* lysates. Randomly selected interacting proteins were analyzed by 10% SDS-PAGE followed by western blot assays, employing the α-EhCP112, α-EhVps23, α-EhVps32 and α-actin antibodies. Input: GST-EhADH. Numbers at the left indicate molecular weight standards in kDa. Arrows signal the immunodetected proteins.

**Figure 4 ijms-25-07609-f004:**
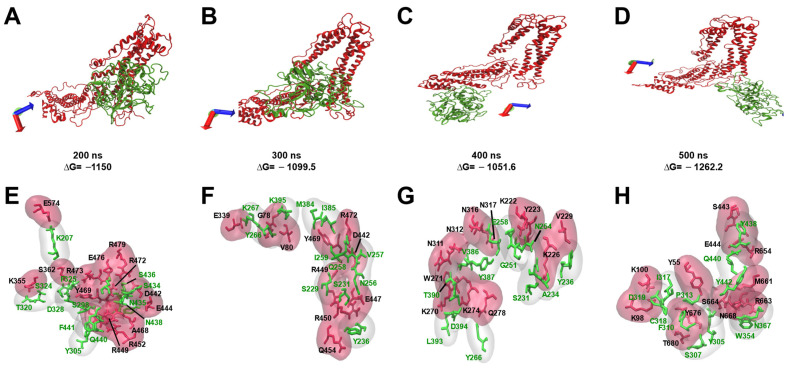
Molecular docking of EhADH and EhCP112. (**A**–**D**) Molecular docking of EhADH with EhCP112 at 200 (**A**), 300 (**B**), 400 (**C**) and 500 (**D**) ns, were obtained using the ClusPro server. EhADH: red. EhCP112: green. ΔG: binding energy. (**E**–**H**) Residues involved in the association of EhADH with EhCP112. Black amino acids belong to EhADH. Green amino acids belong to EhCP112. Axes: X in red, Y in green, Z in blue.

**Figure 5 ijms-25-07609-f005:**
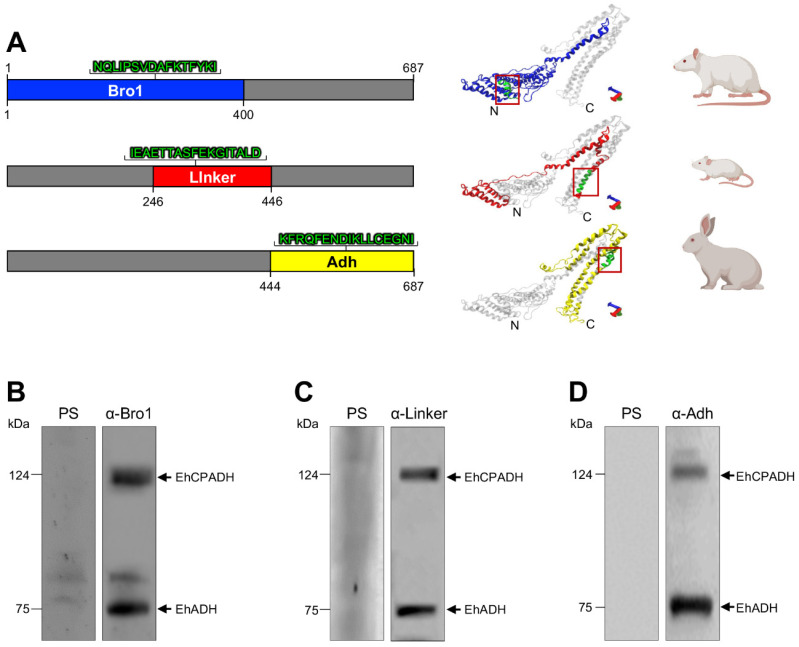
Generation of specific antibodies against the EhADH domains. (**A**) Schematic representation of the primary structure and 3D model of the Bro1 (blue), Linker (red) and Adh (yellow) domains. Sequences displayed in green in the red squares indicate the peptides used for the antibodies’ generation in different animal models. N: amino terminal. C: carboxy terminal. (**B**–**D**) Western blots assays of trophozoite lysates, using the α-Bro1, α-Linker and α-Adh antibodies. PS: Pre-immune sera. Numbers at the left indicate standard molecular weight in kDa. Arrows signal the immunodetected proteins.

**Figure 6 ijms-25-07609-f006:**
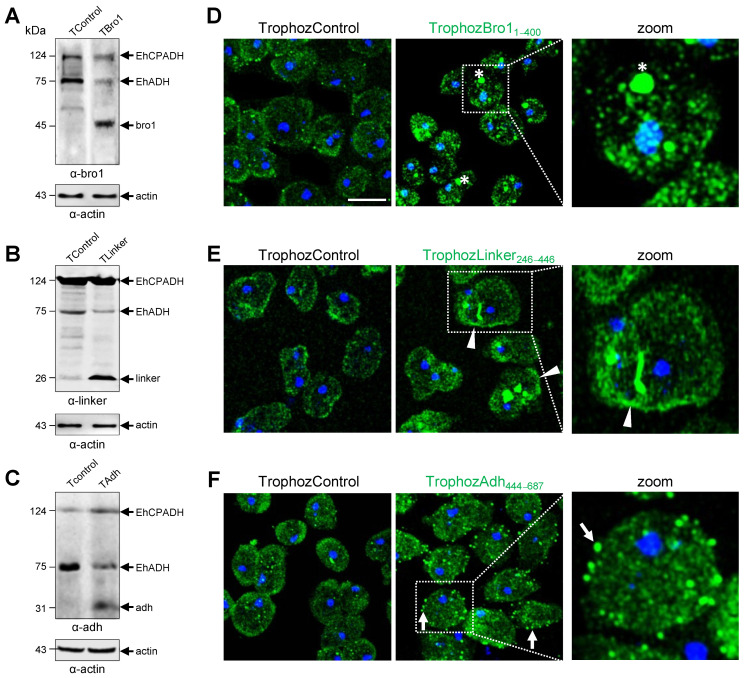
Expression and cellular localization of the EhADH domains in *E. histolytica* trophozoites. TrophozControl, TrophozBro1_1–400,_ TrophozLinker_246–446_ or TrophozAdh_444–687_ were generated by transfection of *pTet*, *pTet/Bro1*, *pTet/Linker* or *pTet/Adh* plasmids, respectively. (**A**–**C**) Western blot assays of trophozoite lysates, using the α-Bro1, α-Linker and α-Adh antibodies. The α-actin antibody was used as a loading control. Numbers at the left indicate standard molecular weight in kDa. Arrows signal the immunodetected proteins. (**D**–**F**) Immunofluorescence experiments of trophozoites employing α-Bro1, α-Linker and α-Adh as primary antibodies and FITC (green) as the secondary antibody. Nuclei (blue) were DAPI stained. Asterisks: cytoplasmic dots that could correspond to vesicular structures; arrowheads: plasma membrane; arrows: dots at cytoplasm and membrane. Bar = 20 μm.

**Figure 7 ijms-25-07609-f007:**
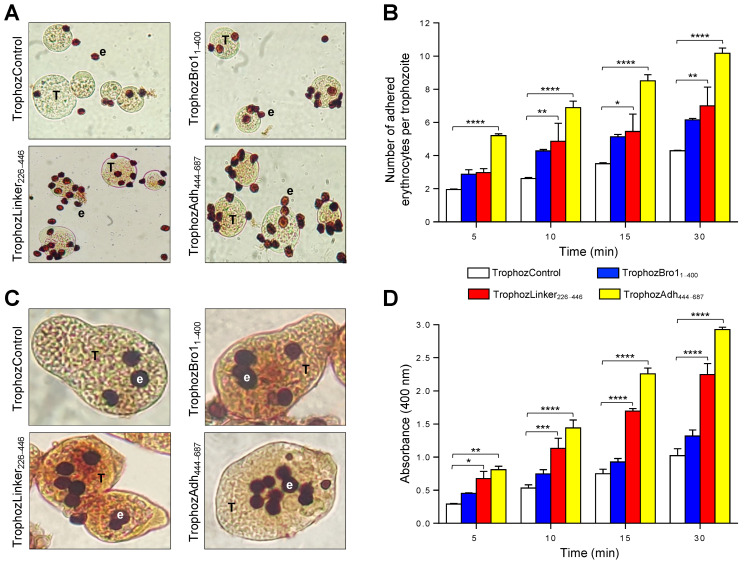
Adhesion to and phagocytosis of erythrocytes by trophozoites overexpressing the EhADH domains. TrophozControl, TrophozBro1_1–400_, TrophozLinker_226–446_ and TrophozAdh_444–687_ populations were incubated at different times with erythrocytes at 4 °C or 37 °C for adhesion and erythrophagocytosis assays, respectively. (**A**) Novikoff staining of trophozoites after 30 min of adhesion assays. (**B**) Number of erythrocytes adhered to trophozoites. Data represent the mean and standard error of the erythrocytes number counted on 100 randomly selected trophozoites in three independent experiments. (**C**) Novikoff staining of trophozoites that ingested erythrocytes for 30 min. T: trophozoite; e: erythrocyte. (**D**) Rate of erythrophagocytosis measured by the hemoglobin concentration inside trophozoites. **** *p* < 0.0001, *** *p* < 0.001 ** *p* < 0.01, * *p* < 0.1.

**Figure 8 ijms-25-07609-f008:**
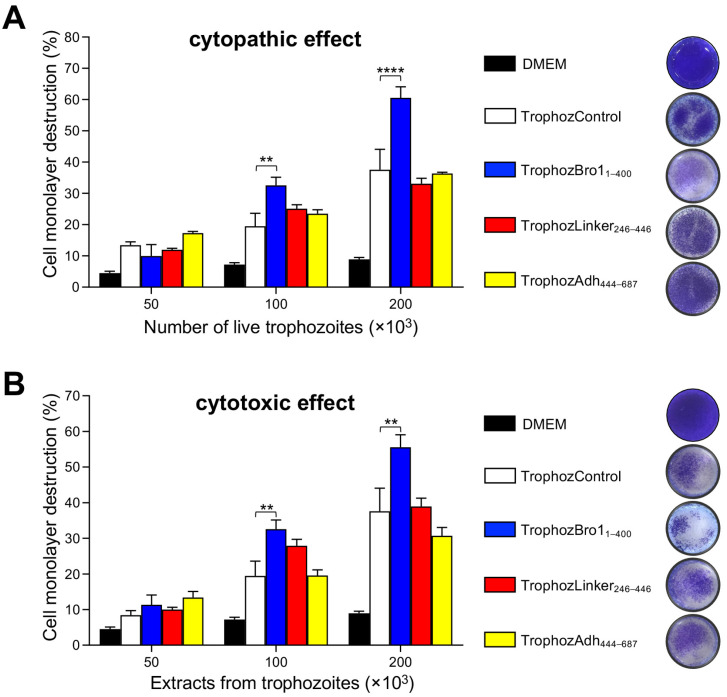
Cytopathic and cytotoxic effect on epithelial cells produced by trophozoites overexpressing the EhADH domains. Destruction of MDCK cell monolayers incubated with live trophozoites (**A**) or extracts from trophozoites (50, 100, and 200 × 10^3^) (**B**). Monolayer damage was spectrophotometrically determined by methylene blue absorbed by the remaining monolayers. Thus, the cellular destruction was deducted from comparing harmed monolayers with confluent cells not incubated with trophozoites (DMEM), which were preserved intact. Representative images of stained MDCK monolayers with 200 × 10^3^ trophozoites are shown at the right. Values represent the mean and standard error of three independent experiments. **** *p* < 0.0001, ** *p* < 0.01.

**Figure 9 ijms-25-07609-f009:**
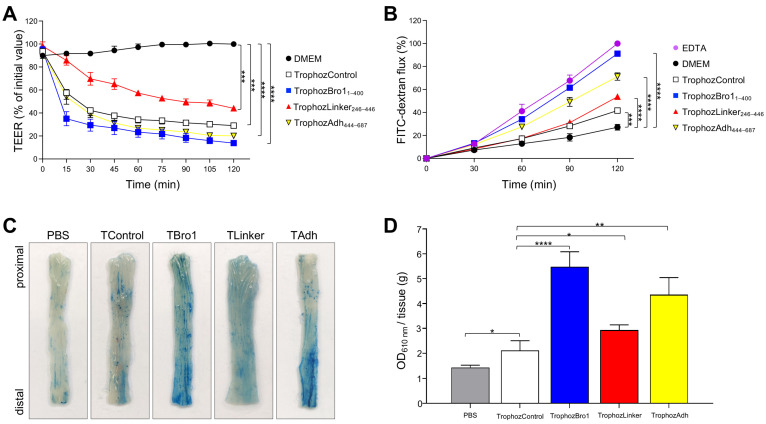
In vitro and in vivo effect of trophozoites overexpressing the EhADH domains on epithelial permeability. TrophozControl, TrophozBro1_1–400_, TrophozLinker_246–446_ and TrophozAdh_444–687_ were incubated with MDCK cells or inoculated in the mice colon. (**A**) MDCK monolayers were incubated with trophozoites for 120 min, and TEER was monitored. TEER was normalized according to the initial value for each Transwell (~1000 Ω·cm^2^). (**B**) FITC-dextran was apically added to MDCK cells grown in Transwells and incubated for 120 min with trophozoites. FITC-dextran obtained from the basal side was measured by fluorescence spectroscopy and normalized according to the epithelial cells treated with 5 mM EDTA used as positive control. DMEM: MDCK cells incubated with culture medium, without parasites. (**C**) C57/BL6 mice were rectally inoculated with trophozoites (10^5^) and after 30 min, the damage in the colonic epithelium was evaluated by Evan’s blue staining absorption. Representative images of distal and proximal portions of the colon after staining are shown. PBS: colon treated with PBS, without parasites. (**D**) Evan’s blue was eluted and spectrophotometrically measured at OD_610nm_ to evaluate the epithelial permeability. Means and standard error are represented for each time point of three independent assays performed in triplicate. **** *p* < 0.0001, *** *p* < 0.001, ** *p* < 0.01, * *p* < 0.1.

**Figure 10 ijms-25-07609-f010:**
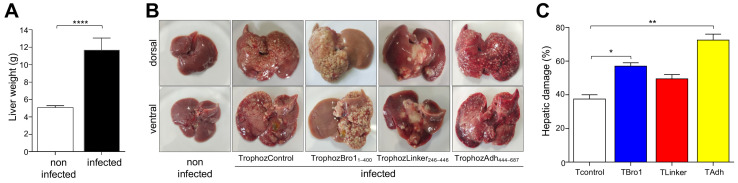
ALA in hamsters inoculated with trophozoites overexpressing the EhADH domains. Hamsters were intraportally inoculated with TrophozControl, TrophozBro1_1–400_, TrophozLinker_246–446_ or TrophozAdh_444–687_ trophozoites (2 × 10^6^). Eight days later, animals were anesthetized, and livers were extracted to examine the size and damage produced. (**A**) Livers´ weights. (**B**) Representative dorsal and ventral views of livers. (**C**) Hepatic damage evaluated as the weight of the abscesses formed divided by the weight of the whole liver before the injured areas were removed. Values represent the mean and standard error of five independent experiments. **** *p* < 0.0001, ** *p* < 0.01, * *p* < 0.1.

**Figure 11 ijms-25-07609-f011:**
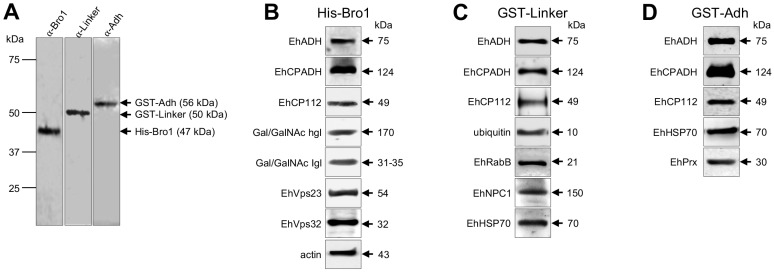
Generation of the recombinant EhADH domains. (**A**) Western blot experiments of recombinant proteins generated from the nucleotide encoding sequence for each domain that was cloned into expression vectors, generating the *pCold/Bro1*, *pGEX/Linker* and *pGEX/Adh* constructs. The expression of recombinant proteins was IPTG-induced. Numbers at the left indicate standard molecular weight in kDa. (**B**–**D**) Western blot assays of interacting proteins obtained from pull-down assays using the recombinant proteins ((**B**): His-Bro1; (**C**): GST-Linker; and (**D**): GST-Adh) and *E. histolytica* lysates. Numbers at the right show the molecular weight of immunodetected proteins.

**Figure 12 ijms-25-07609-f012:**
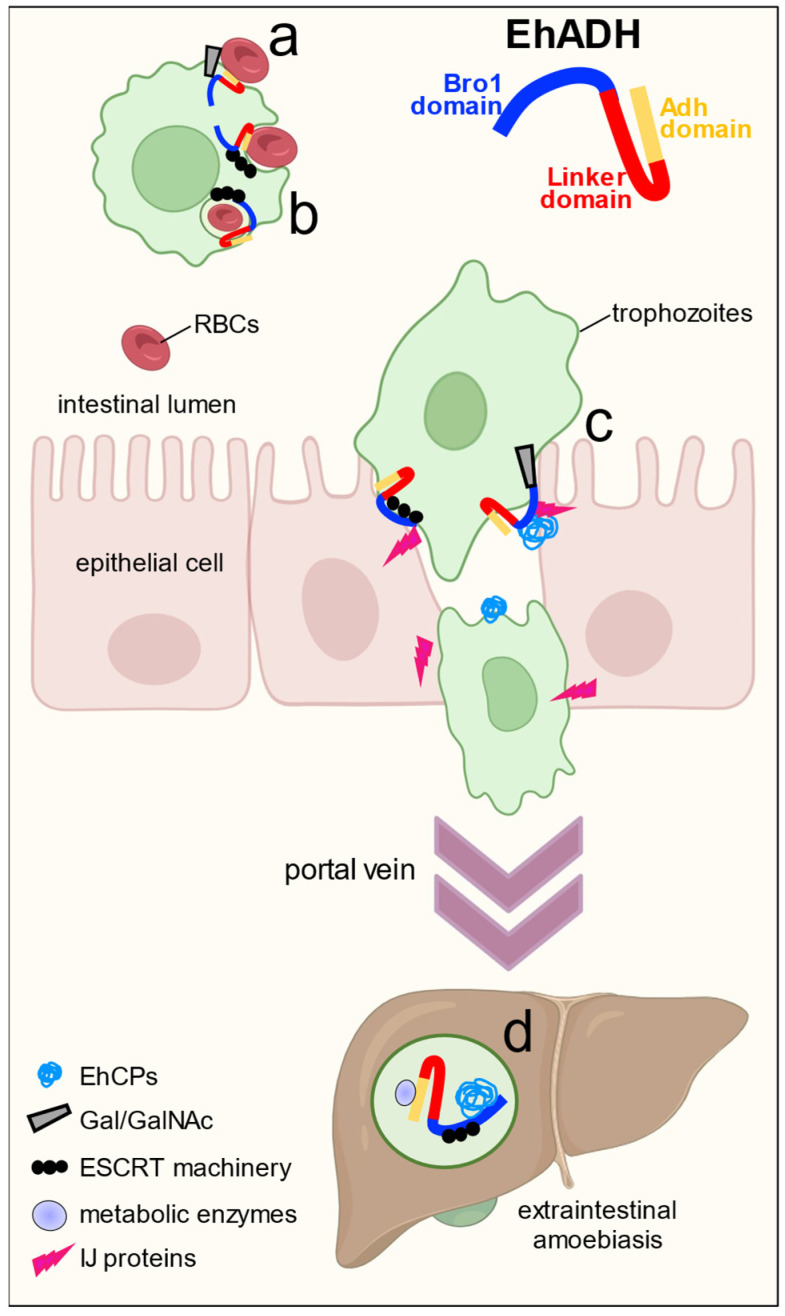
Working model of possible functions for each EhADH domain and recruited molecules involved in virulence events. (a) The Adh domain participates in target cells adhesion, interacting with other parasite molecules including the Gal/GalNAc lectin. (b) It also participates in phagocytosis, together with the ESCRT machinery. (c) Bro1 domain is involved in the IJs disruption, probably by its binging to occludin and claudins, and promotes the complexes formation with EhCPs and ESCRT machinery members. (d) The trophozoites can reach the liver by the portal vein, generating ALA with the main participation of Adh domain, which interplays with EhCPs and metabolic enzymes.

**Table 2 ijms-25-07609-t002:** Mass spectrometry analysis of *E. histolytica* proteins interacting with His-Bro1 and GST-EhADH.

Accession Number	Protein Name		Functions	References
	His-Bro1	GST-EhADH		
EHI_012270 ^†^	Gal/GalNAc lectin heavy subunit *	✗	Adhesion, virulence factor	[46]
EHI_006980 ^†^	Gal/GalNAc lectin Igl1 subunit *	✗	Adhesion, virulence factor	[46]
EHI_181230	EhCP112 *	EhCP112 *	Substrate degradation	[7,25]
EHI_033710 ^†^	EhCP2 *	✗	Substrate degradation	[47,48]
EHI_168240 ^†^	CP *	✗	Substrate degradation	[49]
EHI_178530 ^†^	EhVps23 *	EhVps23 *	Vacuolar transport (ESCRT-I)	[17]
EHI_169820 ^†^	EhVps32 *	EhVps32 *	Vacuolar transport (ESCRT-III)	[19]
EHI_114790	EhVps20 *	✗	Vacuolar transport (ESCRT-III)	[19]
EHI_194400 ^†^	EhVps2 *	EhVps2 *	Vacuolar transport (ESCRT-III)	[19]
EHI_062490 ^†^	EhVps26 *	✗	Retrograde transport (retromer complex)	[50]
EHI_182900 ^†^	Actin *	✗	Cytoskeleton	[27,29]
EHI_008780 ^†^	Actin *	Actin *	Cytoskeleton	[27,29]
EHI_198930 ^†^	Actin *	✗	Cytoskeleton	[27,29]
EHI_049920 ^†^	Tubulin * family protein	✗	Cytoskeleton	[46,51]
EHI_119930 ^†^	Protein kinase domain	✗	Enzyme, phosphate transfer	[22]
EHI_121880 ^†^	Protein kinase domain	✗	Enzyme, phosphate transfer	[22]
EHI_197350	Protein with tyrosine kinase domain	✗	Enzyme, phosphate transfer	[30,31]
EHI_010600 ^†^	BAR/SH3 domain protein	✗	Signaling pathways	[30,31]

The selected proteins had a reliability percentage greater than 95%. * Proteins have been published as related to virulence events. ^†^ Proteins with evidence of expression at membranes or phagosomes in trophozoites, according to AmoebaDB.

**Table 3 ijms-25-07609-t003:** Mass spectrometry analysis of *E. histolytica* proteins interacting with GST-Linker and GST-EhADH.

Accession Number	Protein Name		Functions	References
	His-Bro1	GST-EhADH		
EHI_181220 ^†^	EhADH *	EhADH *	Adhesion	[7,14]
EHI_181230	EhCP112 *	EhCP112 *	Substrate degradation	[7,25]
EHI_166800 ^†^	Ubiquitin *	✗	Protein tag	[22]
EHI_031410 ^†^	GTP-binding protein *	✗	Enzyme (signal transduction, vesicular trafficking, etc.)	[22]
EHI_108610 ^†^	Rab * GTPase	Rab* GTPase	Enzyme (signal transduction, vesicular trafficking, etc.)	[32,34,35]
EHI_164900 ^†^	Rab * GTPase	✗	Enzyme (signal transduction, vesicular trafficking, etc.)	[32,34,35]
EHI_005900	Small Rab7 * GTPase	✗	Enzyme (signal transduction, vesicular trafficking, etc.)	[35,52]
EHI_041950 ^†^	EhVps35 *	✗	Retrograde transport (retromer complex)	[22]
EHI_080220 ^†^	EhNPC1 *	EhNPC1 *	Cholesterol trafficking	[20]
EHI_105240 ^†^	BAR/SH3 domain protein	✗	Signaling pathways	[30,31]
EHI_127700	EhHSP70 *	✗	Chaperone	[46]
EHI_004760 ^†^	Proteosome alpha subunit	✗	Proteosomal	[22]
EHI_152570 ^†^	Ribosomal protein 60S L26	✗	Ribosomal	[22]
EHI_139360	Acetyltransferase, GNAT family	✗	Enzyme	[22,53]
EHI_176700 ^†^	AIG1 family protein	✗	Unknown	[30,31]
EHI_170330	RIO1 family protein	✗	Unknown	[22]

The selected proteins had a reliability percentage greater than 95%. * Proteins have been published as related to virulence events. ^†^ Proteins with evidence of expression at membranes or phagosomes in trophozoites, according to AmoebaDB.

**Table 4 ijms-25-07609-t004:** Mass spectrometry analysis of *E. histolytica* proteins interacting with GST-Adh and GST-EhADH.

Accession Number	Protein Name		Functions	References
	His-Bro1	GST-EhADH		
EHI_181220 ^†^	EhADH *	EhADH *	Adhesion	[7,14]
EHI_061640 ^†^	EhHSP70 *	✗	Chaperone	[46]
EHI_001420 ^†^	EhPeroxiredoxin (EhPrx)	✗	Enzyme	[54,55]
EHI_051060 ^†^	Pyruvate:ferredoxin oxidoreductase	✗	Enzyme	[22,56]
EHI_098570 ^†^	Fructose-1,6-bisphosphate aldolase	✗	Enzyme	[30,31]
EHI_011210 ^†^	Elongation factor 1-Alpha 1	✗	Enzyme	[30,31]
EHI_024240	Aldehyde-alcohol dehydrogenase 2	✗	Enzyme	[22,57]
EHI_051940 ^†^	Poly (ADP-ribose) glucohydrolase	✗	Enzyme	[30,31]
EHI_152650 ^†^	Flavoprotein type A	✗	Enzyme	[30,31]
EHI_044970 ^†^	Malic enzyme	✗	Enzyme	[30,31]
EHI_068660 ^†^	60S ribosomal protein L5	✗	Ribosomal	[22]
EHI_050130 ^†^	60S ribosomal protein L14	✗	Ribosomal	[22]
EHI_035600 ^†^	Ribosomal protein L18a	✗	Ribosomal	[22]
EHI_057670 ^†^	20 kDa antigen	✗	Antigen	[30,31]

The selected proteins had a reliability percentage greater than 95%. * Proteins have been published as related to virulence events. ^†^ Proteins with evidence of expression at membranes or phagosomes in trophozoites, according to AmoebaDB.

## Data Availability

Data is contained within the article or Appendix A.

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
