# Peer review of "Entamoeba histolytica: EhADH, an Alix Protein, Participates in Several Virulence Events through Its Different Domains"

_ijms, 2024, doi:10.3390/ijms25147609_

Round 1
Reviewer 1 Report
Comments and Suggestions for Authors
The authors present an interesting study where they report on an investigation into the role of EhADH in the adhesion and virulence of Entamoeba histolytica.
The study materials and methods are well described and would enable replication of the study.
The results are well presented though I have made some suggestions for the authors to consider. Similarly, the results support the conclusions that have been drawn.
Overall the manuscript is well written, though it is quite wordy in places. Many of the results sections start by repeating some of the background details and some consideration should be given to deleting these paragraphs.
I would strongly suggest that the authors provide the primary western blotting images as supplemental data clearly linking the images used in the manuscript to these. This is fast becoming the standard approach to ensure there are no questions about the images presented in the manuscript. I am NOT suggesting there has been any inappropriate manipulation etc with the images in this manuscript.
One element that is lacking from the discussion is around limitation(s) of the current study. I would suggest that the authors consider this in the revision of their manuscript. As an example, the authors have identified various proteins that interact with the domains of EhADH. However, it is plausible that steric hinderances of the domains when in the whole molecule might prevent the binding of some of the potential partners that bind the expressed domains. Few studies are perfect and answer all the questions, so it is important to consider what the limitations are.
I would suggest the authors add a paragraph to the end of the discussion that specifically addresses the stated aims of the study.
General comments and suggestions.
Line 2 The current title is very general and could be revised to be more specific concerning the putative roles of the EhADH domains in Entamoeba histolytica pathogenesis.
Line 15 I do not understand the meaning of the sentence below, please review and revise for clarity.
“EhADH follows the ingested pray since the first contact until its arrival to the multivesicular bodies.”
Line 17 The abbreviation “ESCRT” should be explained in full.
Line 17 I would suggest modifying this sentence to make it clear that these domains make up EhADH, consider:
“Here, we produced trophozoites overexpressing the three domains of EhADH, Bro1 (1-400 aa), Linker (246-446 aa) and Adh (444-687 aa) to evaluate their role in virulence.”
I have also suggested replacing “define” with “evaluate”, as very little in biology is definitive. I think the authors need to develop a meaningful nomenclature for their modified trophozoites, that carries through the manuscript. Currently, there are multiple terms describing the same mutants which makes it difficult to follow the story. Consider the following:
“Here, we produced trophozoites overexpressing the three domains of EhADH, Bro1 (1-400 aa, TrophozΔBro1), Linker (246-446 aa, TrophozΔLink) and Adh (444-687 aa, TrophozΔAdh) to evaluate their role in virulence.”
These terms could then be used throughout the manuscript to clearly identify what mutant is being described.
Line 18 suggest revision “The TrophozΔBro1 indicated slightly increased”
Line 65-66 suggest revision:
Thus, the EhADH primary contribution to E. histolytica virulence makes it necessary to dissect the domains that potentially facilitate interactions with other molecules to form the complexes involved in epithelial lysis and phagocytosis.
Line 68 suggest replacing “regions” with “domains”, I think it best use the same terms throughout the manuscript.
Lines 80-85 This text, Figure 1 and Figure 2A read more like introductory text and could fit seamlessly with the text on Line 52 – consider moving.
Line 124 I would suggest using a superscript to signify these proteins rather than bold text to ensure they are clearly identifiable.
Lines 203 to 209 & Line 220-231 This text describes the results for the RBC binding of transfected trophozoites over time, illustrated in Figure 5. The text refers to percentages, however none of the panels in Figure 5 show percentages. As the figures are provided to support the interpretations and conclusions drawn, it is essential that the text and figures report the data consistently.
I would suggest that the text be modified to describe the data as shown in Figure 5.
Line 250 Figure 6. As panels A and B show the effects of live and lysed trophozoites respectively, given these are different treatments, I think the titles on the graphs
How do the representative images of the MDCK monolayers correlate to the treatments shown on the x-axes of the graphs?
Were the extracts of the trophozoites used to generate Fig 6B quantified in any way, for example, mg per mL of protein?
Perhaps the x-axes titles could be modified to:
Fig 6A Number of live trophozoites (×105)
Fig 6B Extracts from trophozoites (×105)
Or similar to ensure it is clear that the treatments are different.
Line 393 I would suggest using a superscript to signify these proteins rather than bold text to ensure they are clearly identifiable. Same comment for any subsequent tables showing similar data.
Line 392 Table 2, Line 403 Table 3, Line 412 Table 4, I would suggest adding another column to each of these tables that indicates of the putative interacting partners for each domain that were shown to interact with the complete EhADH protein, as shown in Table 1.
Comments on the Quality of English Language
See comments to authors.
Author Response
Comments and Suggestions for Authors
The authors present an interesting study where they report on an investigation into the role of EhADH in the adhesion and virulence of Entamoeba histolytica.
The study materials and methods are well described and would enable replication of the study.
The results are well presented though I have made some suggestions for the authors to consider. Similarly, the results support the conclusions that have been drawn.
Comment 1: Overall the manuscript is well written, though it is quite wordy in places. Many of the results sections start by repeating some of the background details and some consideration should be given to deleting these paragraphs.
I would strongly suggest that the authors provide the primary western blotting images as supplemental data clearly linking the images used in the manuscript to these. This is fast becoming the standard approach to ensure there are no questions about the images presented in the manuscript. I am NOT suggesting there has been any inappropriate manipulation etc with the images in this manuscript.
One element that is lacking from the discussion is around limitation(s) of the current study. I would suggest that the authors consider this in the revision of their manuscript. As an example, the authors have identified various proteins that interact with the domains of EhADH. However, it is plausible that steric hinderances of the domains when in the whole molecule might prevent the binding of some of the potential partners that bind the expressed domains. Few studies are perfect and answer all the questions, so it is important to consider what the limitations are.
I would suggest the authors add a paragraph to the end of the discussion that specifically addresses the stated aims of the study.
Response 1: Dear reviewer we sincerely appreciate your work to improve our manuscript.
The background details that are repetitive were deleted in the Results section.
Furthermore, the primary western blotting images were previously uploaded during the submission for the editorial office, but we also have attached the file for this reviewer. However, if you still considering, we can include the information in the supplementary section.
In the discussion section, a limitation sentence about steric hinderances has been included (line 560-565).
Following your suggestion about the addition of a paragraph to the end of the discussion that specifically addresses the stated aims of the study, and attending the request from the reviewer 2, we have added a Conclusion section, where we have highlighted the aim of our paper to resume the goal and findings of this work.
Comment 2: General comments and suggestions.
Line 2 The current title is very general and could be revised to be more specific concerning the putative roles of the EhADH domains in Entamoeba histolytica pathogenesis.
Response 2: Following your pertinent suggestion, we changed the title of this article for:
“Entamoeba histolytica: EhAdh, an Alix protein, participates in several virulence events through its different domains”
Comment 3: Line 15 I do not understand the meaning of the sentence below, please review and revise for clarity.
“EhADH follows the ingested pray since the first contact until its arrival to the multivesicular bodies.”
Response 3: The sentence was changed for:
“EhADH adheres to the pray and follows it through its arrival to the multivesicular bodies”
Comment 4: Line 17 The abbreviation “ESCRT” should be explained in full.
Response 4: It was explained in lines 15-16.
Comment 5: Line 17 I would suggest modifying this sentence to make it clear that these domains make up EhADH, consider:
“Here, we produced trophozoites overexpressing the three domains of EhADH, Bro1 (1-400 aa), Linker (246-446 aa) and Adh (444-687 aa) to evaluate their role in virulence.”
Response 5: Thank you very much, it was considered and changed.
Comment 6: I have also suggested replacing “define” with “evaluate”, as very little in biology is definitive. I think the authors need to develop a meaningful nomenclature for their modified trophozoites, that carries through the manuscript. Currently, there are multiple terms describing the same mutants which makes it difficult to follow the story. Consider the following:
“Here, we produced trophozoites overexpressing the three domains of EhADH, Bro1 (1-400 aa, TrophozΔBro1), Linker (246-446 aa, TrophozΔLink) and Adh (444-687 aa, TrophozΔAdh) to evaluate their role in virulence.”
These terms could then be used throughout the manuscript to clearly identify what mutant is being described.
Line 18 suggest revision “The TrophozΔBro1 indicated slightly increased”
Response 6: Thank you very much for your excellent comment. However, we believe that is clearer in the following way, what do you think?
TrophozControl, TrophozBro11-400, TrophozLinker246-446 and TrophozAdh444-687.
Comment 7: Line 65-66 suggest revision:
Thus, the EhADH primary contribution to E. histolytica virulence makes it necessary to dissect the domains that potentially facilitate interactions with other molecules to form the complexes involved in epithelial lysis and phagocytosis.
Response 7: In order to improve the meaning of this sentence, it was changed for:
“Due to the EhADH's seminal contribution on E. histolytica virulence, the aim of our work was to dissect the domains to facilitate their potential interactions with other molecules to form complexes involved in epithelial lysis and phagocytosis” (lines 71-73)
Comment 8: Line 68 suggest replacing “regions” with “domains”, I think it best use the same terms throughout the manuscript.
Response 8: Following your suggestion, we have changed it.
Comment 9: Lines 80-85 This text, Figure 1 and Figure 2A read more like introductory text and could fit seamlessly with the text on Line 52 – consider moving.
Response 9: The figure 1 was moved to line 56.
We also included a new figure 2 containing the 3D EhADH structure and a chart of the workflow to help readers to understanding the methodological approach, as the reviewer 2 have suggested.
Comment 10: Line 124 I would suggest using a superscript to signify these proteins rather than bold text to ensure they are clearly identifiable.
Response 10: Following your suggestion, bolded proteins were changed for an asterisk in all tables. The notation is below each table.
Comment 11: Lines 203 to 209 & Line 220-231 This text describes the results for the RBC binding of transfected trophozoites over time, illustrated in Figure 5. The text refers to percentages, however none of the panels in Figure 5 show percentages. As the figures are provided to support the interpretations and conclusions drawn, it is essential that the text and figures report the data consistently.
I would suggest that the text be modified to describe the data as shown in Figure 5.
Response 11: The number of adhered of phagocyted erythrocytes were added (together with percentage) in the text as shown in Figure 5.
Comment 12: Line 250 Figure 6. As panels A and B show the effects of live and lysed trophozoites respectively, given these are different treatments, I think the titles on the graphs
How do the representative images of the MDCK monolayers correlate to the treatments shown on the x-axes of the graphs?
Response 12: The images representing the damage of MDCK cells incubated with 200,000 live trophozoites or extracts from trophozoites, which were spectrophotometrically measured, calculating the remaining monolayer in each well. Thus, the cellular destruction was deducted from comparing harmed monolayers with confluent cells incubated only with DMEM medium, which were preserved intact. Images and data were included in the figure 7 as DMEM condition. The damage calculation was explained in the figure legend and in the Materials and Methods section.
Comment 13: Were the extracts of the trophozoites used to generate Fig 6B quantified in any way, for example, mg per mL of protein?
Perhaps the x-axes titles could be modified to:
Fig 6A Number of live trophozoites (×105)
Fig 6B Extracts from trophozoites (×105)
Or similar to ensure it is clear that the treatments are different.
Response 13: The x-axes titles in figure 6 were changed following your suggestion. Number of live trophozoites and Extract from trophozoites. In addition, we have written the title of each graph.
Comment 14: Line 393 I would suggest using a superscript to signify these proteins rather than bold text to ensure they are clearly identifiable. Same comment for any subsequent tables showing similar data.
Response 14: The significant proteins were noted with asterisks in all tables instead of bold letters.
Comment 15: Line 392 Table 2, Line 403 Table 3, Line 412 Table 4, I would suggest adding another column to each of these tables that indicates of the putative interacting partners for each domain that were shown to interact with the complete EhADH protein, as shown in Table 1.
Response 15: The column was added attending your suggestion.

Reviewer 2 Report
Comments and Suggestions for Authors
The authors should define clearly the objectives of their study. Please include the general work frame for this study and also please describe the working hypothesis for the study.
At the end, was the hypothesis confirmed or refuted?
Rest of Introduction is OK.
Materials and methods. In this section, please give details of all controls used in the study: control chemicals, control methodologies, etc. All these must be include in a separate sub-section that will be add therein, in order to provide details about controls. Did you use another protozoan organism as control pathogen?
Also, I suggest to include a chart of the workflow to help readers in understanding the methodological approach.
Analysis. Before using the statistical test employed, please provide evidence about the normality of distribution of the data. In a supplementary table, please give the full details of all dataset assessed for normality: please provide the following for each dataset: W, sum of squares, b, skewness, skewness shape, excess kyrtosis, kyrtosis shape, outliers.
After confirmation of normality, the statistical tests will be accepted. Otherwise, the analysis should be redone by means of non-parametric tests.
The tables with the primers must be moved to supplementary material.
Results.
Comment about visualization. The graphs must be colourised please.
With regard to tables, possibly some of the results can be presented in new tables to be added in the revised version with respective reduction of text in the manuscript.
Discussion.
There are some relevant references regarding similar studies in other protozoa, which have been omitted. So, please do include the results of previous researchers and please discuss them versus yours.
Please, no figures in Discussion.
Please provide a brief passage regarding possible commercialization of the findings and potential applications of the findings in clinical settings. Do you think that we shall be able to apply the findings in patients within 2024 or 2025? What is your proposed timeframe for clinical applications of the findings?
Conclusions.
Missing, so please add the relevant section.
Overall. Manuscript that can advance to next stage after extensive revision.
Author Response
Reviewer 2
Comments and Suggestions for Authors
Comment 1: The authors should define clearly the objectives of their study. Please include the general work frame for this study and also please describe the working hypothesis for the study.
At the end, was the hypothesis confirmed or refuted?
Response 1: We appreciate your opinion on our work, and following your right suggestions we clarified the general objective of this manuscript, a sentence was added in the Abstract, Discussion and Conclusion sections.
Introduction section: “Due to the EhADH's seminal contribution on E. histolytica virulence, the aim of our work was to dissect the domains to facilitate their potential interactions with other molecules to form complexes involved in epithelial lysis and phagocytosis” (lines 71-73).
The hypothesis of our work was that the multiple EhADH functions lies on its different domains, mainly in the Bro1 and the Adh domains (lines 73-75).
We have added in the Conclusion section a paragraph that indicate that our hypothesis was confirmed (lines 586-590).
Rest of Introduction is OK.
Comment 2: Materials and methods. In this section, please give details of all controls used in the study: control chemicals, control methodologies, etc. All these must be included in a separate sub-section that will be add therein, in order to provide details about controls. Did you use another protozoan organism as control pathogen?
Response 2: Following your suggestions, we have included the sub-section “4.1. Chemical and Reagents” in the Materials and Methods section to specify the high purity and reliability of substances used in this work. Necessary controls are punctually mentioned and resumed for each one of the experiments performed.
When recombinant proteins were used, the GST- or His-tag proteins were included as controls. Similarly, when trophozoites mutants were employed, trophozoites transfected with the empty vector were used to compare results. In the pull-down assays, the resins (Glutathione-Sepharose and cobalt beads) were applied also as controls (lines 713-714).
In our group Alix proteins has been identified in other Entamoeba species by bioinformatics approaches; however, they are not been characterized in those organisms. So far, in the many years that we have worked with E. histolytica, we have not used other protozoa as control. Here, we neither consider performing the experiments with other organisms. However, we will analyze your suggestion for further studies.
Comment 3: Also, I suggest to include a chart of the workflow to help readers in understanding the methodological approach.
Response 3: Following your pertinent suggestion, we have included a chart of the workflow in the new figure 2B,C.
Comment 4: Analysis. Before using the statistical test employed, please provide evidence about the normality of distribution of the data. In a supplementary table, please give the full details of all dataset assessed for normality: please provide the following for each dataset: W, sum of squares, b, skewness, skewness shape, excess kyrtosis, kyrtosis shape, outliers.
After confirmation of normality, the statistical tests will be accepted. Otherwise, the analysis should be redone by means of non-parametric tests.
Response 4: Eight tables and graphs containing all these data have been included in the supplementary section.
Comment 5: The tables with the primers must be moved to supplementary material.
Response 5: The table was moved to supplementary material.
Comment 6: Results.
Comment about visualization. The graphs must be colourised please.
Response 6: Attending your request, graphs were colorized.
Comment 7: With regard to tables, possibly some of the results can be presented in new tables to be added in the revised version with respective reduction of text in the manuscript.
Response 7: The relevant information was already incorporated in the tables 1-4; however, we included a column in the tables 2-4 to compare the findings between partners domains and partners-whole protein.
Comment 8: Discussion.
There are some relevant references regarding similar studies in other protozoa, which have been omitted. So, please do include the results of previous researchers and please discuss them versus yours.
Response 8: In our group other Alix proteins has been identified in other protozoa parasites by bioinformatics approaches; however, they are not been studied or characterized in those organisms, so far. We have performed a new search and we could not find Alix proteins in protozoan parasites. However, we found Alix in Dictyostelium discoideum and in Echinococcus multilocularis that have been used markers in extracellular vesicles and in development of the organism. Both references have been included in Discussion section (lines 489-494).
Comment 9: Please, no figures in Discussion.
Response 9: We have moved Figure 11 (now figure 4) to the Results section (lines 146-155). However, we believe that the model to explain and summarize our work must have a place in the Discussion section. We hope that it will be fine for you.
Comment 10: Please provide a brief passage regarding possible commercialization of the findings and potential applications of the findings in clinical settings. Do you think that we shall be able to apply the findings in patients within 2024 or 2025? What is your proposed timeframe for clinical applications of the findings?
Response 10: Thank you for your excellent suggestion. Indeed, we are working to upgrade our results to generate a vaccine against amoebiasis, but the way is long still.
In the Conclusion section a sentence regarding this concern was added:
“We are convinced that this molecule can be used as a potential candidate to develop therapeutical approaches or vaccines in the future”.
Comment 11: Conclusions
Missing, so please add the relevant section.
Response 11: Conclusion section was included.
Overall. Manuscript that can advance to next stage after extensive revision.
Reviewer 3 Report
Comments and Suggestions for Authors
This manuscript from Zanatta et al details experiments elucidating the contributions of three segments of the Entamoeba histolytica protein EhADH to its functionality. Entamoeba histolytica causes gastrointestinal amoebiasis in humans, a disease which results in more than 55,000 yearly deaths worldwide, highlighting the need for the development of a vaccine and further anti-protozoan treatments. When Entamoeba histolytica enters the gastrointestinal tract (and other tissues such as the liver in severe systemic cases), it adheres to and lyses epithelial cell barriers, resulting in cell death, destruction of extracellular complexes such as tight junctions and the formation of tissue lesions. EhADH is a protein that comprises part of the EhCPADH complex known to be critical for cell adherence and lysis. EhADH contains a N-terminal Bro1 domain consistent with the ALIX protein family known for regulating cell death and a C-terminal region with a unique "V" structure containing the EhADH adhesion domain (Adh). However, further work was needed to characterize the contributions of each domain to in-vitro and in-vivo behavior of Entamoeba histolytica infection.
By overexpressing Bro-1 and Adh (as well as a middle section that also contained phosphorylation sites from Bro-1 termed Linker) in trophozoites, the authors determined that Bro1 was most implicated in epithelial barrier destruction in cell culture and in mouse colon, while Adh was most implicated in red blood cell adhesion and phagocytosis, and interestingly also in liver damage in the progression of intestinal to systemic infection in hamsters. Linker had more intermediate or more mild phenotypes. The authors hypothesize these difference in functions might be aided by different localizations and binding partners, as Bro-1 was found throughout the cytoplasm and was observed to bind with Gal/GalNac lectin, ESCRT pathway proteins, actin and tubulin, while Adh may be more heavily located at the plasma membrane.
As a whole, the paper is well structured and straightforward, and the new studies on contributions of each domain to in vivo pathogenesis in particular were interesting new contributions to understanding the functions of EdADH. I have the following minor comments to be addressed prior to acceptance for publication:
1) The authors should clarify how long overexpression was carried out for prior to using trophozoites for experiments, and include data on cellular vitality of overexpressing vs control trophozoites across this time period.
2) While the large FITC+ sections of Bro1 in figure 4D suggest that it may co-localize with cytoplasmic vesicles, the authors can't definitely conclude this without also staining for markers of vesicles or including a citation where similar morphology is an acceptable marker for vesicles.
3) Given the different gross pathologies observed in figure 8b (decreased blood flow with Bro-1 trophozoites and necrotic areas for Adh), the authors should include histological staining for livers shown in this figure.
4) Relating to line 457, the authors should comment on any known literature on EhADH oligomerization and whether any functional differences in EhADH domains could be hypothesized (since they were all found by mass spec to bind with EhADH).
Author Response
Reviewer 3
Comments and Suggestions for Authors
This manuscript from Zanatta et al details experiments elucidating the contributions of three segments of the Entamoeba histolytica protein EhADH to its functionality. Entamoeba histolytica causes gastrointestinal amoebiasis in humans, a disease which results in more than 55,000 yearly deaths worldwide, highlighting the need for the development of a vaccine and further anti-protozoan treatments. When Entamoeba histolytica enters the gastrointestinal tract (and other tissues such as the liver in severe systemic cases), it adheres to and lyses epithelial cell barriers, resulting in cell death, destruction of extracellular complexes such as tight junctions and the formation of tissue lesions. EhADH is a protein that comprises part of the EhCPADH complex known to be critical for cell adherence and lysis. EhADH contains a N-terminal Bro1 domain consistent with the ALIX protein family known for regulating cell death and a C-terminal region with a unique "V" structure containing the EhADH adhesion domain (Adh). However, further work was needed to characterize the contributions of each domain to in-vitro and in-vivo behavior of Entamoeba histolytica infection.
By overexpressing Bro-1 and Adh (as well as a middle section that also contained phosphorylation sites from Bro-1 termed Linker) in trophozoites, the authors determined that Bro1 was most implicated in epithelial barrier destruction in cell culture and in mouse colon, while Adh was most implicated in red blood cell adhesion and phagocytosis, and interestingly also in liver damage in the progression of intestinal to systemic infection in hamsters. Linker had more intermediate or more mild phenotypes. The authors hypothesize these difference in functions might be aided by different localizations and binding partners, as Bro-1 was found throughout the cytoplasm and was observed to bind with Gal/GalNac lectin, ESCRT pathway proteins, actin and tubulin, while Adh may be more heavily located at the plasma membrane.
As a whole, the paper is well structured and straightforward, and the new studies on contributions of each domain to in vivo pathogenesis in particular were interesting new contributions to understanding the functions of EdADH. I have the following minor comments to be addressed prior to acceptance for publication:
Comment 1) The authors should clarify how long overexpression was carried out for prior to using trophozoites for experiments, and include data on cellular vitality of overexpressing vs control trophozoites across this time period.
Response 1: We sincerely appreciate the opinion on our work of this Reviewer.
The overexpression experiments performed with an inducible vector (pEhHYG-tetR-O-CAT), and the expression of EhADH domains was achieved applying tetracycline for 24 h as detailed in Material and Methods section. We include the cellular growth graph of transfected trophozoites.
The cellular growth of overexpressing vs control trophozoites was monitored during 7 days, and not significant differences were observed among different populations as described in supplementary figure 1. The cells were frozen, however, the induction with G-418 takes several days to stabilize again the transfected cells.
Comment 2) While the large FITC+ sections of Bro1 in figure 4D suggest that it may co-localize with cytoplasmic vesicles, the authors can't definitely conclude this without also staining for markers of vesicles or including a citation where similar morphology is an acceptable marker for vesicles.
Response 2: We agree with you that we cannot affirm that detected structures are bone fide vesicles without the use of specific markers in the figure 4D (now, figure 5D); therefore, we changed the expression for: “cytoplasmic dots that could correspond to vesicular structures” (lines 208-209).
Comment 3) Given the different gross pathologies observed in figure 8b (decreased blood flow with Bro-1 trophozoites and necrotic areas for Adh), the authors should include histological staining for livers shown in this figure.
Response 3: We appreciate very much your excellent suggestion and for sure we will do these studies later. However, we think for this paper the hepatic change is really evident and in this study. Furthermore, the generation of the ALA in the animal model implies at least three months, because we need to defrozen and stabilize the tranfected trophozoites and the performed the experiment in animal with appropriate age. After this, we would obtain the livers for the histological examination. Our experimets were carried out at least three times.
Comment 4) Relating to line 457, the authors should comment on any known literature on EhADH oligomerization and whether any functional differences in EhADH domains could be hypothesized (since they were all found by mass spec to bind with EhADH).
It was mentioned in the lines 511-512 and the references were included:
Response 4: “as has been described for other Alix proteins, which are capable to form multimers (18, 69-71)”
- Bissig C, Gruenberg J. ALIX and the multivesicular endosome: ALIX in Wonderland. Trends in Cell Biology. 2014. pp. 19–25. doi:10.1016/j.tcb.2013.10.009
- Mercier V, Laporte MH, Destaing O, Blot B, Blouin CM, Pernet-Gallay K, et al. ALG-2 interacting protein-X (Alix) is essential for clathrin-independent endocytosis and signaling. Sci Rep. 2016;6. doi:10.1038/srep26986
- Pires R, Hartlieb B, Signor L, Schoehn G, Lata S, Roessle M, et al. A Crescent-Shaped ALIX Dimer Targets ESCRT-III CHMP4 Filaments. Structure. 2009;17. doi:10.1016/j.str.2009.04.007
- Bissig C, Lenoir M, Velluz MC, Kufareva I, Abagyan R, Overduin M, et al. Viral Infection Controlled by a Calcium-Dependent Lipid-Binding Module in ALIX. Dev Cell. 2013;25. doi:10.1016/j.devcel.2013.04.003
Round 2
Reviewer 2 Report
Comments and Suggestions for Authors
All the issues were addressed. I have no more comments.